# Regulation of B cell fate by chronic activity of the IgE B cell receptor

Zhiyong Yang[1,2], Marcus J Robinson[1,2†], Xiangjun Chen[3†], Geoffrey A Smith[4], Jack Taunton[4], Wanli Liu[3], Christopher D C Allen[1,2,5*]

[1]Cardiovascular Research Institute, University of California, San Francisco, San Francisco, United States; [2]Sandler Asthma Basic Research Center, University of California, San Francisco, San Francisco, United States; [3]MOE Key Laboratory of Protein Sciences, Collaborative Innovation Center for Diagnosis and Treatment of Infectious Diseases, School of Life Sciences, Institute for Immunology, Tsinghua University, Beijing, China; [4]Department of Cellular and Molecular Pharmacology, University of California, San Francisco, San Francisco, United States; [5]Department of Anatomy, University of California, San Francisco, San Francisco, United States

**Abstract** IgE can trigger potent allergic responses, yet the mechanisms regulating IgE production are poorly understood. Here we reveal that IgE[+] B cells are constrained by chronic activity of the IgE B cell receptor (BCR). In the absence of cognate antigen, the IgE BCR promoted terminal differentiation of B cells into plasma cells (PCs) under cell culture conditions mimicking T cell help. This antigen-independent PC differentiation involved multiple IgE domains and Syk, CD19, BLNK, Btk, and IRF4. Disruption of BCR signaling in mice led to consistently exaggerated IgE[+] germinal center (GC) B cell but variably increased PC responses. We were unable to confirm reports that the IgE BCR directly promoted intrinsic apoptosis. Instead, IgE[+] GC B cells exhibited poor antigen presentation and prolonged cell cycles, suggesting reduced competition for T cell help. We propose that chronic BCR activity and access to T cell help play critical roles in regulating IgE responses.

*For correspondence: Chris. Allen@ucsf.edu

†These authors contributed equally to this work

Competing interests: The authors declare that no competing interests exist.

## Introduction

Of all the antibody isotypes, immunoglobulin E (IgE) can elicit the most rapid immune responses in immediate-type hypersensitivity, contributing to the pathogenesis of numerous allergic diseases (*Gould et al., 2003*). IgE-mediated hypersensitivity responses are typically localized to specific tissues such as the skin, nose, lung, or intestine, whereas systemic responses can result in life-threatening anaphylaxis. Only a small fraction of individuals with allergic diseases will experience anaphylaxis, however, suggesting that IgE responses are normally restricted. Indeed, IgE is the least abundant antibody isotype in serum. While the availability of IgE in serum is limited in part by a short half-life and binding to Fc receptors, the production of secreted IgE appears to be tightly regulated (*Geha et al., 2003*).

In order to understand the regulation of IgE production, recent attention has focused on IgE-expressing (IgE[+]) B cells. Little had been known about these cells due to their low abundance and technical difficulties in their detection. These challenges have largely been overcome by the generation of fluorescent IgE reporter mice as well as improved technical methods (*He et al., 2013*; *Talay et al., 2012a*; *Wesemann et al., 2011*; *Yang et al., 2012*). Initial studies of IgE[+] B cells in mice have revealed several key differences from B cells expressing IgG1, the other major isotype induced in type 2 immune responses.

**eLife digest** Antibodies are proteins that recognize and bind to specific molecules, and so help the immune system to defend the body against foreign substances that are potentially harmful. In some cases, harmless substances – such as pollen, dust or food – can trigger this response and lead to an allergic reaction. A type of antibody called immunoglobulin E (IgE) is particularly likely to trigger an allergic response.

In general, immune cells called plasma cells produce antibodies and release them into the body. However, in B cells – the cells from which plasma cells develop – the antibodies remain on the surface of the cells. Here, the antibody acts as a "receptor" that allows the B cell to tell when its antibody has bound to a specific substance.

Generally, B cells only activate when their B cell receptors bind to a specific substance. This binding triggers signals inside the cell that determine its fate – such as whether it will develop into a plasma cell. Recent studies have shown that B cells that have IgE on their surface (IgE$^+$ B cells) are predisposed to develop rapidly into plasma cells.

To investigate why this is the case, Yang et al. have now studied B cells both in cell culture and in mice. The results show that the IgE B cell receptor autonomously signals to the cell even when it is not bound to a specific substance, in a manner that differs from other types of B cell receptors. This increases the likelihood that the IgE$^+$ B cell will develop into a plasma cell and limits the competitive fitness of IgE$^+$ B cells. These findings provide new insights into how IgE responses are regulated by the B cell receptor.

The next step will be to determine, at a molecular level, the basis for the autonomous signaling produced by the IgE B cell receptor when it is not bound to a specific substance. It will then be possible to investigate how this mechanism compares with the way that signals are normally transmitted when a B cell receptor binds to a specific substance.

IgE$^+$ B cells appeared only transiently and at low frequencies in germinal centers (GCs) (*He et al., 2013*; *Talay et al., 2012b*; *Yang et al., 2012*). These structures form during immune responses in lymphoid tissues and are major sites for antibody affinity maturation as well as the generation of long-lived plasma cells (PCs) and memory B cells (*Allen et al., 2007a*; *Victora and Nussenzweig, 2012*). Consistent with the limited participation of IgE$^+$ B cells in GCs, IgE$^+$ responses typically exhibit reduced affinity maturation compared with IgG1$^+$ responses, and most of the affinity maturation that does occur requires an IgG1$^+$ B cell intermediate (*Erazo et al., 2007*; *He et al., 2013*; *Xiong et al., 2012*; *Yang et al., 2012*). In addition, we observed a relative paucity of long-lived IgE$^+$ PCs (*Yang et al., 2012*) and other groups reported that memory IgE responses were largely initiated by non-IgE-expressing B cells (*He et al., 2013*; *Katona et al., 1991*; *Turqueti-Neves et al., 2015*). Taken together, it appears that IgE$^+$ B cells undergo an abortive GC phase that limits downstream IgE responses.

In contrast, a larger proportion of IgE$^+$ B cells were observed to have a PC phenotype compared with IgG1$^+$ B cells in several studies (note that here we use a broad definition of PCs to refer to all antibody secreting cells, including plasmablasts) (*Erazo et al., 2007*; *He et al., 2013*; *Laffleur et al., 2015*; *Yang et al., 2012*). We reported that this observation could be recapitulated in cell culture of primary mouse B cells, suggesting the increased PC differentiation of IgE$^+$ B cells was B cell intrinsic (*Yang et al., 2012*). We proposed that the propensity of IgE$^+$ B cells to undergo terminal differentiation into PCs may directly contribute to the low frequency and disappearance of IgE$^+$ B cells from GCs (*Yang et al., 2012*). Another group proposed that IgE$^+$ GC B cells exhibit diminished BCR signaling and undergo increased rates of apoptosis (*He et al., 2013*).

Both intrinsic and extrinsic mechanisms could account for the distinct features of IgE$^+$ B cells. A likely candidate for intrinsic regulation is the expression of the IgE B cell receptor (BCR). Each isotype of BCR has a different constant region sequence which may confer different signaling capabilities. IgG BCRs can promote enhanced responses compared with IgM BCRs, most notably enhanced PC differentiation in recall responses (*Martin and Goodnow, 2002*). This is thought to be due, at least in part, to the extended intracellular cytoplasmic tail of the IgG BCR, compared with the short

three amino acid sequence (KVK) in IgM (*Martin and Goodnow, 2002*). A conserved tyrosine motif in the cytoplasmic tail of the IgG BCR, which was reported to be primarily responsible for its differential signaling, is also present in the IgE BCR (*Engels et al., 2009*). The cytoplasmic tail of the IgE BCR has also been implicated in promoting apoptosis through binding the mitochondrial protein Hax1 (*Laffleur et al., 2015*).

Here we show that the IgE BCR is a major determinant of IgE$^+$ B cell fate. In the presence of T cell help signals, the IgE BCR promoted PC differentiation. This cell fate predisposition occurred in the absence of cognate antigen, whereas the IgG1 BCR promoted PC differentiation only in the presence of cognate antigen. Multiple domains of the BCRs were responsible for the difference between the IgE and IgG1 isotypes. The propensity of the IgE BCR to induce antigen-independent PC differentiation was associated with a weak, constitutive activity of the IgE BCR. Genetic or pharmacological perturbations in BCR signaling led to reduced PC differentiation of IgE$^+$ B cells in the absence of cognate antigen in cell culture. In immunized mice, reductions in BCR signaling led to consistently increased IgE$^+$ GC B cell responses with variable effects on PCs. The effects of BCR signaling on IgE$^+$ GC B cell responses could not be explained by differential rates of apoptosis, as we found no evidence that the IgE BCR directly promotes apoptosis. Instead, BCR signaling slowed the cell cycle progression of IgE$^+$ GC B cells, and low IgE BCR expression limited antigen uptake and presentation. Thus, IgE B cell responses are restrained by a predisposition toward early PC differentiation, prolonged cell cycles, and limited access to T cell help, leading to reduced affinity maturation and memory cell generation.

## Results

### The IgE BCR promotes antigen-independent PC differentiation

Several studies in mice have observed that a larger fraction of IgE$^+$ cells have a PC phenotype compared with IgG1$^+$ cells after immunization (*Erazo et al., 2007*; *He et al., 2013*; *Yang et al., 2012*). We previously reported that this in vivo observation could be recapitulated in primary B cell cultures with anti-CD40 antibodies and IL-4 (*Yang et al., 2012*), which promote class switch recombination (CSR) to IgE and IgG1. In these B cell cultures, a substantially greater fraction of IgE$^+$ cells were PCs compared with IgG1$^+$ cells. This result was confirmed again here using CD138 (Syndecan-1) as a marker of PCs (*Figure 1A and B*). We found that a larger fraction of IgE$^+$ cells than IgG1$^+$ cells had a PC phenotype regardless of the concentration of anti-CD40 antibody, although we noted that stronger CD40 stimulation was inhibitory toward PC differentiation (*Figure 1A and B*). We also previously established that the increased PC differentiation occurs in IgE$^+$ B cells that have undergone the same number of cell divisions as IgG1$^+$ B cells (*Yang et al., 2012*). These observations established a strong correlation between CSR to IgE and PC differentiation, but the cause of this correlation was unknown. Notably, these culture conditions mimic T cell help but do not stimulate the BCR. We hypothesized that the IgE BCR itself promotes PC differentiation in the absence of cognate antigen. We sought to test this model by ectopically expressing the IgE BCR in primary B cells.

In order to determine whether the IgE BCR directly promoted PC differentiation, we developed an approach to ectopically express the IgE BCR versus other BCR isotypes. We chose retroviral transduction, which is a robust method to express genes in primary B cells (*Wu et al., 1987*). However, initial experiments with standard retroviral vectors were hampered by variable expression. We therefore engineered a retroviral vector using the EF1α promoter (*Figure 1C*, *Figure 1—figure supplement 1*), which we found gave much more uniform and robust expression. A heavy chain variable region specific for 2,4,6-trinitrophenyl (TNP) was linked to the heavy chain constant regions of each BCR isotype (*Figure 1C*). As a reporter of transduction, Cerulean (*Rizzo et al., 2004*), a derivative of cyan fluorescent protein, was placed upstream of the heavy chain, linked by a 2A sequence (*de Felipe et al., 2006*) which allows translationally-linked expression of multiple proteins from a single transcript (*Figure 1C* and *Figure 1—figure supplement 1*).

With this optimized retroviral vector, we expressed BCRs of various isotypes in B cells from AID-deficient (*Aicda$^{-/-}$*) mice that cannot undergo class switch recombination (*Muramatsu et al., 2000*). B cell proliferation was again stimulated with anti-CD40 and IL-4. Transduced cells were identified as Cerulean$^+$ and PC differentiation was measured by the expression of the marker CD138 (Syndecan-1). Strikingly, among all isotypes tested, the IgE BCR promoted the highest frequency of PC

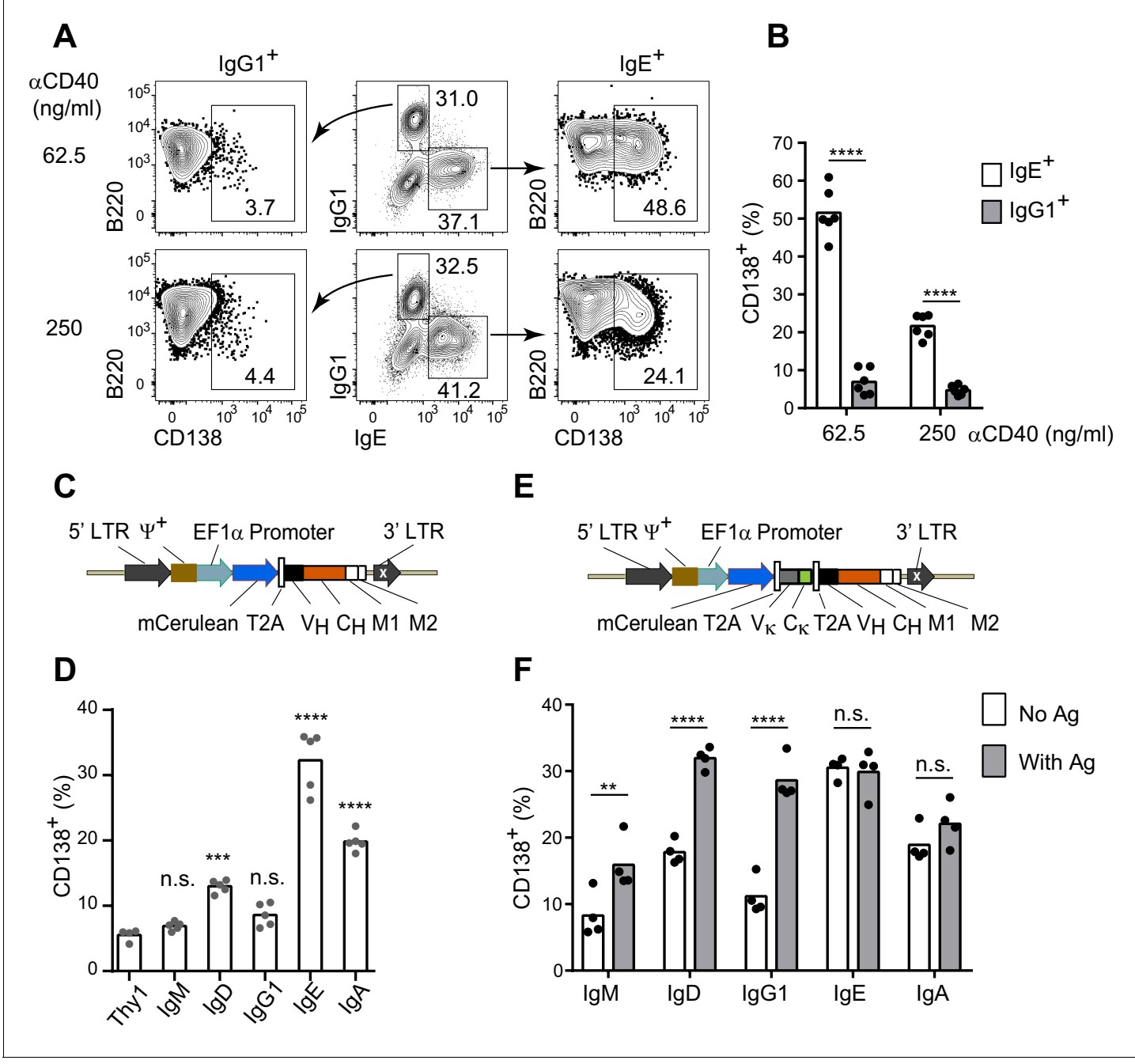

**Figure 1.** The IgE BCR promotes antigen-independent PC differentiation of mouse B cells in culture. (**A** and **B**) Representative flow cytometry (**A**) and quantification (**B**) of PC differentiation (CD138+) from wild-type B cells cultured for 4 d with IL-4 and anti-CD40 (αCD40). Cells were pre-gated as IgD−IgM− and with a broad B220+ gate. (**C**) Diagram of the retroviral construct for the ectopic expression of BCR heavy chains of various isotypes. A detailed vector diagram is provided in *Figure 1—figure supplement 1*. (**D**) Frequency of PCs (CD138+) among AID-deficient B cells ectopically expressing different BCR isotypes or Thy1.1 (Thy1) as a control. Cells were cultured for a total of 4 d with anti-CD40 and IL-4 and were retrovirally transduced on d 1. (**E**) Diagram of the retroviral construct for the ectopic expression of both heavy and light chains of BCRs specific for TNP. (**F**) Frequency of PCs (CD138+) among AID-deficient B cells ectopically expressing different isotypes of TNP-specific BCRs in the absence or presence of TNP-OVA antigen (Ag). Cells were cultured for 4d with anti-CD40 and IL-4, were retrovirally transduced on d 1, and antigen was added on d 2. Similar data were obtained with B1-8^flox/+ Cγ1^Cre/+ B cells (*Figure 1—figure supplement 2*). Transduced cells for (**D**) and (**F**) were identified as Cerulean+. Dots represent data points from individual experiments. Bars represent the mean. LTR, long terminal repeat, which on the 3' end is self-inactivating (white x); ψ+, extended packaging signal; V_H and V_κ, coding sequences for the variable region of the heavy and light chains, respectively; C_H and C_κ, coding sequences for constant regions of the heavy chain and light chain, respectively; M1, M2, coding sequences for the M1 and M2 exons; n.s., not

*Figure 1 continued on next page*

*Figure 1 continued*

significant; **p<0.01; ***p<0.001; ****p<0.0001 (t-tests with the Holm-Sidak correction for multiple comparisons (B,F), or one-way ANOVA with Dunnett's post-test comparing each heavy chain to the Thy1.1 control sample (D)).
The following figure supplements are available for figure 1:

**Figure supplement 1.** Vector for retroviral transduction of BCRs.
**Figure supplement 2.** Antigen-independent versus antigen-dependent PC differentiation with B1-8$^{flox/+}$ C$\gamma$1$^{Cre/+}$ B cells.

differentiation in the absence of cognate antigen (*Figure 1D*). The IgM and IgG1 BCRs did not promote PC differentiation in the absence of cognate antigen, whereas the IgD and IgA BCRs had intermediate effects (*Figure 1D*). Since the IgG1 BCR has been reported to be much more efficient than IgM at inducing PC differentiation in vivo (*Martin and Goodnow, 2002*), we modified our system to test antigen-dependent effects. Specifically, we generated retroviral constructs encoding both the light chain and heavy chain from a TNP-specific monoclonal antibody, in which the heavy chain variable region was linked to the constant regions of various heavy chain isotypes (*Figure 1E*). These TNP-specific BCR constructs were then transduced into AID-deficient B cells. Upon the addition of TNP-ovalbumin (OVA), the TNP-specific IgG1 BCR promoted robust PC differentiation, similar to the IgE BCR (*Figure 1F*). The addition of TNP-OVA also resulted in a more moderate increase in PC differentiation in cells transduced with TNP-specific IgM BCRs, with intermediate results with IgD BCRs (*Figure 1F*). However, the addition of TNP-OVA did not cause a further increase in PC differentiation in cells transduced with TNP-specific IgE or IgA BCRs (*Figure 1F*). Taken together, these data suggest that the IgE BCR is a strong inducer of PC differentiation in an antigen-independent manner, mimicking the behavior of an antigen-engaged IgG1 BCR.

## Multiple domains of the IgE BCR contribute to antigen-independent PC differentiation

Having established that the IgE BCR promotes antigen-independent PC differentiation which can be recapitulated by ectopic expression, we sought to determine which domain(s) of IgE were responsible for this activity by domain swap experiments with IgG1. Initial efforts focused on the intracellular cytoplasmic tail (CT) region, which is thought to be responsible for major differences in signaling among BCR isotypes (*Wienands and Engels, 2016*). Surprisingly, antigen-independent PC differentiation was unaffected when the IgE CT was replaced with that of IgG1 (*Figure 2A*). The transmembrane (TM) region of the BCR is thought to mediate association with the signaling adapters Ig$\alpha$ and Ig$\beta$ (*Reth, 1992*), yet swapping the TM region also had no impact on antigen-independent PC differentiation (*Figure 2A*). Membrane BCRs also contain a short extracellular segment proximal to the membrane that is unique to each isotype, known as the membrane Ig isotype-specific (migis) segment or the extracellular membrane proximal domain (*Davis et al., 1991*; *Major et al., 1996*). The expression of constructs in which the migis of IgE was swapped with that of IgG1 led to a profound loss of antigen-independent PC differentiation (*Figure 2A*). In reverse swaps, in which these domains of IgE were introduced into IgG1, the migis enhanced antigen-independent PC differentiation, but more striking results were seen when the IgE migis was combined with the IgE TM and CT, but not with the TM alone, suggesting that the CT may also be able to contribute to antigen-independent PC differentiation (*Figure 2A*). Taken together, the IgE migis region appears to be necessary, but not sufficient, for antigen-independent PC differentiation mediated by the IgE BCR. Full antigen-independent activity of the IgE BCR required the IgE migis region to be combined with either the extracellular domains of IgE or the CT of IgE.

In the course of these experiments, we also measured the surface expression of IgM, which remains genetically encoded in the AID-deficient B cells that we transduced with IgE versus IgG1 BCRs. We observed that IgM was downmodulated in B cells transduced with IgE (*Figure 2B*, construct 1) but not with IgG1 (*Figure 2B*, construct 3). The IgM downmodulation was dependent on the migis region, as revealed by transducing constructs with the migis regions swapped (*Figure 2B*, constructs 2 and 4). Indeed, in cells transduced with each of the constructs shown in *Figure 2A*, all of the constructs that contained the IgE migis resulted in downmodulation of surface IgM, whereas

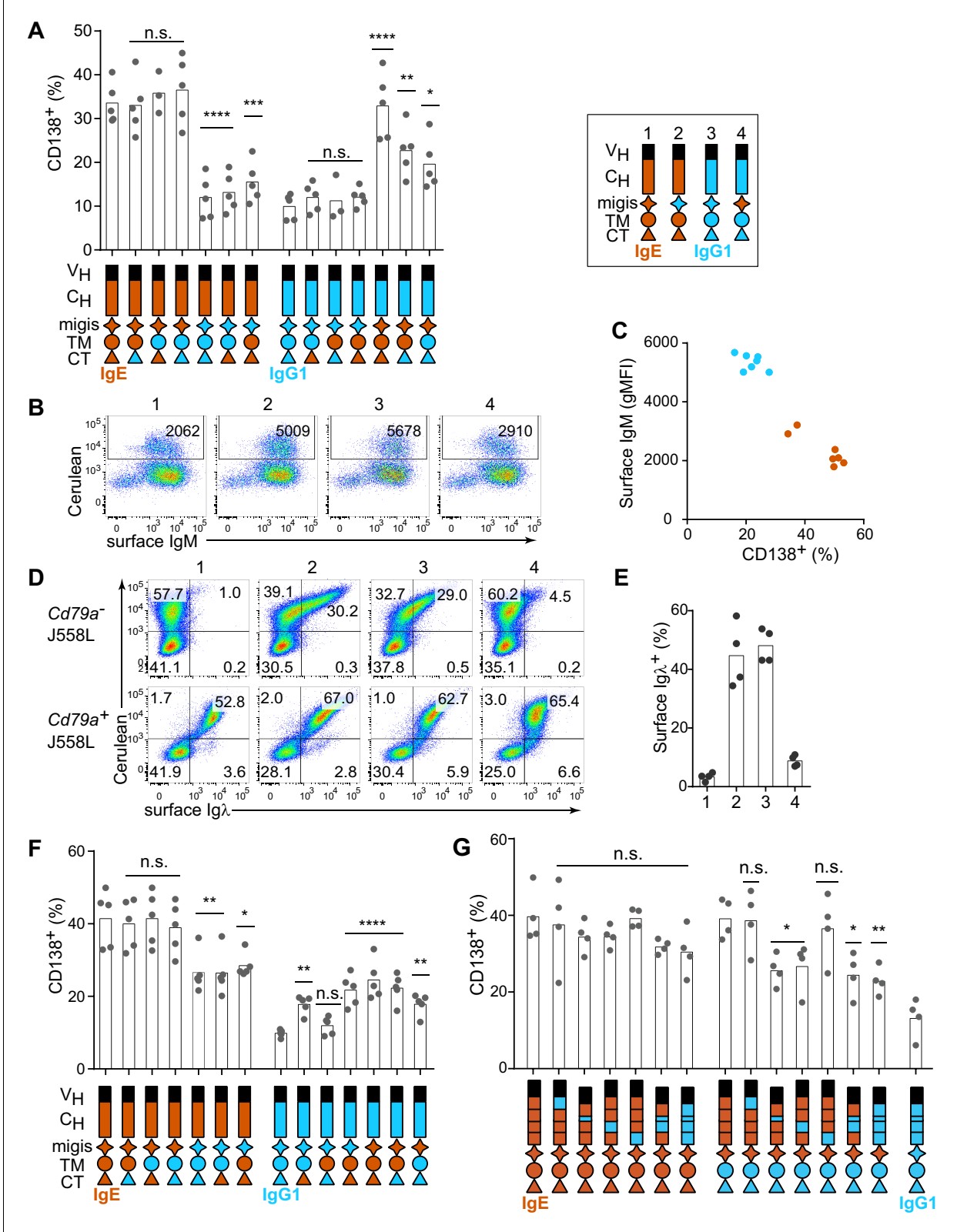

**Figure 2.** Contribution of different domains of the IgE BCR to antigen-independent PC differentiation. Cells were retrovirally transduced with constructs in which the domains of IgE (orange) and IgG1 (blue) were swapped as illustrated (see legend in upper-right for numbered constructs in (B, D, and E)). Primary B cells were cultured with anti-CD40 and IL-4 in (A–C and F–G). (A) Frequency of PC differentiation (CD138+) among transduced AID-deficient B cells. (B) Representative flow cytometry of the surface abundance of IgM in transduced AID-deficient B cells. Numbers in the plots show the gMFI of

*Figure 2 continued on next page*

Figure 2 continued

surface IgM. (C) Inverse correlation of cell surface IgM with PC differentiation (CD138$^+$) in transduced AID-deficient B cells. Each dot represents the results from B cells transduced with a distinct chimeric BCR from the constructs shown in (A); dots in orange represent chimeric BCRs with the migis derived from IgE; dots in blue represent chimeric BCRs with the migis derived from IgG1. (D) Representative flow cytometry of surface BCR (λ light chain) expression on transduced J558L cells. The lower panels depict cells that had been stably transduced with *Cd79a* (Igα). (E) Quantification of surface BCR (λ light chain) expression on transduced (Cerulean$^+$) J558L cells (*Cd79a$^-$*). (F and G) Frequency of PC differentiation (CD138$^+$) among transduced B1-8$^{flox/+}$ Cγ1$^{Cre/+}$ B cells. In (G) the individual CH domains of IgE and IgG1 were swapped as illustrated. Surface BCR and Cerulean reporter expression are provided in *Figure 2—figure supplement 1*. Transduced cells were identified as Cerulean$^+$ (A–E) with the addition of IgM$^-$IgD$^-$ (F–G). Dots represent data points from individual experiments, except in (C) where the data are from a single experiment representative of three independent experiments. Bars represent the mean. V$_H$, coding sequence for the variable region of the heavy chain; C$_H$, coding sequence for the constant region of the heavy chain; TM, transmembrane region; CT, cytoplasmic tail; gMFI, geometric mean fluorescence intensity; n.s., not significant. **p<0.01, ***p<0.001, ****p<0.0001 (for each group of related constructs, one-way ANOVA with Dunnett's post-test comparing each construct to the leftmost parent construct).

The following figure supplement is available for figure 2:

**Figure supplement 1.** Surface expression of BCRs on naturally class-switched versus transduced primary B cells.

the constructs that contained the IgG1 migis did not (*Figure 2C*). We hypothesized that this IgM downmodulation might reflect competition of the transduced BCRs versus IgM for binding to Igα. It was reported that similar to IgM, the IgE BCR depends on binding to Igα for export from the endoplasmic reticulum, whereas the IgG BCR does not (*Venkitaraman et al., 1991*). This difference among isotypes had been attributed to the TM domain (*Reth, 1992*; *Venkitaraman et al., 1991*), but we considered whether the migis, which is proximal to the membrane, could also be involved in Igα association. We transduced the IgE versus IgG1 BCRs, or constructs in which the migis regions were swapped, into J558L cells, which lack Igα (*Hombach et al., 1990*). For comparison, we transduced the BCRs into J558L cells in which we had stably transduced Igα. Remarkably, the IgE migis made BCR surface expression completely dependent on Igα, whereas the IgG1 migis permitted substantial surface BCR localization in the absence of Igα (*Figure 2D and E*). These data indicate that the migis region is a major site of interaction with Igα.

In order to determine whether residual IgM expression in AID-deficient B cells was affecting our analysis of IgE versus IgG1 BCR domains, we repeated our experiments in B cells in which the pre-existing IgM BCR is deleted, analogous to natural CSR. Specifically, we made use of mice with a loxP-flanked B1-8 Ig heavy chain variable region allele (B1-8$^{flox}$), which could be deleted with Cre recombinase (*Lam et al., 1997*). These mice were bred to mice carrying Cγ1 (*Ighg1*)-Cre (*Casola et al., 2006*), which is efficiently induced by anti-CD40 and IL-4, resulting in Cre-mediated deletion of the existing B1-8$^{flox}$ BCRs. We retrovirally transduced these cells with new BCRs close in time to the deletion of the existing BCR, mimicking natural CSR. Ectopic expression of BCRs of different isotypes in B1-8$^{flox/+}$ Cγ1$^{Cre/+}$ B cells gave similar results to AID-deficient B cells, with IgE promoting a high frequency of antigen-independent PC differentiation, whereas IgG1 promoted antigen-dependent PC differentiation (*Figure 1—figure supplement 2*). In domain swap experiments in B1-8$^{flox/+}$ Cγ1$^{Cre/+}$ cells, the IgE migis region still contributed to antigen-independent PC differentiation, but played a less prominent role than in the AID-deficient B cells, presumably since no IgM was present to compete for Igα (*Figure 2F*). Specifically, in IgE BCR constructs with the IgG1 migis, antigen-independent PC differentiation was reduced compared with constructs with the IgE migis, but was still elevated compared with the full IgG1 BCR, suggesting only a partial requirement of the migis region for antigen-independent PC differentiation. In reverse domain swaps in which regions of the IgG1 BCR were substituted with their counterparts in IgE, we observed a synergistic contribution of both the IgE migis and IgE CT to antigen-independent PC differentiation (*Figure 2F*). However, when the IgG1 extracellular domains were coupled with the IgE migis, IgE TM, and IgE CT, the frequency of antigen-independent PC differentiation was intermediate between that of the full IgE BCR and IgG1 BCRs (*Figure 2F*), again suggesting a contribution of the IgE extracellular domains, which were further explored below. Taken together, these data indicated that the IgE migis and CT both contributed to, but could not fully account for, the antigen-independent activity of the IgE BCR.

We therefore tested the contribution of the extracellular domains of the IgE BCR to antigen-independent PC differentiation. The IgE BCR has four extracellular constant region domains (CH1-4), whereas the IgG1 BCR has three domains (CH1-3), with the second domain (CH2) of IgE replaced by a hinge in IgG1 (*Gould et al., 2003*). As expected, in the context of the IgE migis and CT, which we established above were major contributors to antigen-independent PC differentiation, there was no significant effect of swapping the extracellular domains, although there was a trend suggesting a contribution of the IgE CH2 and CH3 domains (*Figure 2G*). We therefore considered whether a contribution of the extracellular domains could be further revealed in hybrid constructs containing the IgG1 CT, to remove the contribution of the IgE CT. Indeed, both the IgE CH2 and IgE CH3 reproducibly contributed to antigen-independent PC differentiation, as revealed by transducing hybrid constructs in which these domains had been swapped with the IgG1 hinge and CH2, respectively (*Figure 2G*). Thus, multiple parts of the IgE molecule, specifically the CH2, CH3, migis, and CT, all contribute to antigen-independent PC differentiation, making this BCR distinct from the IgG1 BCR.

In order to further validate the results from our isotype and domain swap BCR comparisons, we measured the surface expression of the ectopically-expressed BCRs. The surface expression of the transduced IgE and IgG1 BCRs were equivalent to each other, as measured with an antibody to the light chain which pairs with these heavy chains (*Figure 2—figure supplement 1A*). We also compared the surface expression of the transduced BCRs with the endogenous BCRs in normal B cells that had been induced to undergo natural CSR to IgE versus IgG1. We observed that the surface expression of the transduced BCRs was overlapping with the surface expression of normal endogenous IgE BCRs, but less than that of normal endogenous IgG1 BCRs (*Figure 2—figure supplement 1A*). This difference was due to the fact that membrane IgE normally has lower expression than membrane IgG1 (*Figure 2—figure supplement 1A*), as previously reported (*He et al., 2013*). Therefore, in the transduction system we did not 'overexpress' the BCRs but rather achieved a surface abundance similar to normal membrane IgE. We also noted that the induction of PC differentiation by the IgE BCR occurred over the entire range of surface BCR expression, indicating that small changes in surface expression would be unlikely to impact our results (*Figure 2—figure supplement 1B*). All domain swap constructs had equivalent expression of the Cerulean reporter and achieved measurable surface IgE and/or IgG1 expression within approximately a 4-fold range (*Figure 2—figure supplement 1C*). We therefore conclude that our system allowed a fair comparison of IgE versus IgG1 BCR domains for the ability to promote antigen-independent PC differentiation.

## Constitutive activity of the IgE BCR

We next sought to determine whether the antigen-independent PC differentiation mediated by IgE BCR was due to antigen-independent BCR signaling that differed from the IgG1 BCR. Initial attempts to look at phosphorylation of the downstream signaling adapters such as Syk, Btk, Erk, and Akt, by phosflow failed to show striking differences between IgE$^+$ and IgG1$^+$ B cells (data not shown), presumably because many of these phosphorylation events are transient and can only be observed within minutes of strong acute stimulation, whereas the antigen-independent activity may be weaker and constitutive. We therefore considered cumulative readouts of BCR activity. We found that a larger fraction of IgE$^+$ B cells than IgG1$^+$ B cells expressed the activation marker CD69 (*Figure 3A*), which was particularly apparent with low concentrations of anti-CD40 antibody, since strong CD40 stimulation could itself promote CD69 upregulation (data not shown). Recently, it has been reported that Nur77 upregulation is a readout that is very sensitive to antigen-receptor signaling but only weakly induced by CD40 stimulation (*Zikherman et al., 2012*). IgE$^+$ B cells had higher constitutive Nur77 expression than IgG1$^+$ B cells, as revealed by a Nur77-GFP reporter (*Figure 3B*). The elevated CD69 and Nur77 expression suggest that IgE$^+$ B cells have higher constitutive BCR activity than IgG1$^+$ B cells.

To gain further insights into the constitutive activity of the IgE versus IgG1 BCRs, we imaged these BCRs on the plasma membrane of quiescent B cells by total internal reflection fluorescence (TIRF) microscopy. BCR clustering on the surface of B cells is thought to be important for BCR signaling after antigen activation as well as for tonic BCR signaling (*Pierce and Liu, 2010*), which is essential for B cell survival (*Kraus et al., 2004*; *Lam et al., 1997*). We visualized BCR clustering through yellow fluorescent protein (YFP)-tagged Igα, which was expressed in J558L cells together with either IgE or IgG1 BCRs. J558L cells expressing similar amounts of IgE and IgG1 both showed high fluorescence intensity BCR cluster spots on the plasma membrane (*Figure 3C*), consistent with previous

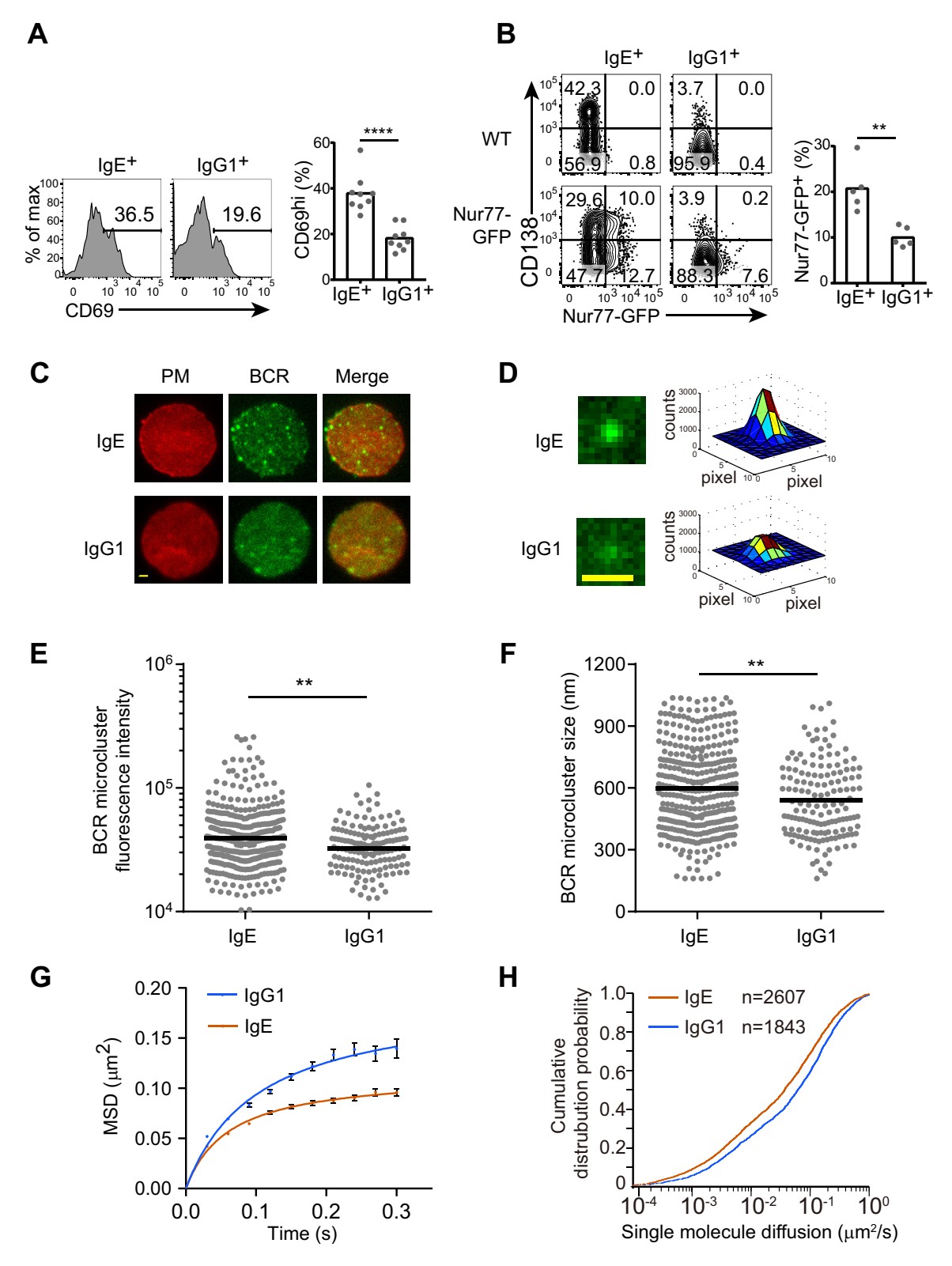

**Figure 3.** The IgE BCR exhibits differential constitutive activity compared with the IgG1 BCR. (A and B) Primary B cells were cultured for 4 d with IL-4 and anti-CD40 followed by flow cytometric evaluation of IgE⁺ and IgG1⁺ cells as in *Figure 1A*. Representative flow cytometry (left) and quantification (right) of the frequency of cells that were CD69⁺ (A) or Nur77-GFP⁺ (B). Dots represent individual samples pooled from four (A) or five (B) experiments. (C–H) TIRF microscopy of J558L cells that were transduced with the IgE or IgG1 BCRs together with Igα-YFP ('BCR') and stained to show

*Figure 3 continued on next page*

Figure 3 continued

the plasma membrane (PM). (C and D). Representative TIRF microcopy images of single cells (C) and individual BCR clusters (D). Scale bars, 1.5 μm. (E and F) Quantification of the fluorescence intensity (E) and size (F) of BCR microclusters. Dots indicate individual measurements. (G and H) Characterization of the Brownian diffusion coefficient of IgE and IgG1 BCRs by single molecule tracking, displayed as mean squared displacement (MSD) versus time (G) and cumulative probability distribution (H) plots. Bars show the mean (A, B, and F) or the geometric mean (E). Error bars (G) indicate the SEM. **p<0.01, ***p<0.001, ****p<0.0001 (Mann-Whitney U-test).

reports that a minor fraction of BCRs are clustered to support tonic signaling in resting B cells (*Pierce and Liu, 2010*). However, some of the IgE BCR clusters were noticeably higher in fluorescence intensity (*Figures 3C, D and E*) and tended to be larger in size (*Figure 3F*) than the IgG1 BCR clusters. These results suggest that IgE BCRs intrinsically form more prominent clusters compared with IgG1 BCRs in quiescent B cells.

The mobility feature of BCRs is another dimension which may indicate the status of receptors and downstream signaling. We used single molecule tracking to quantify the diffusion rate of IgE or IgG1 BCR molecules on the surface of J558L cells. This tracking analysis illuminated that the Brownian diffusion of IgE BCRs was more confined compared to that of IgG1 BCRs (*Figure 3G*). Moreover, the short-range diffusion coefficients of single BCR molecules were processed and derived into a cumulative distribution probability plot, which confirmed that IgE BCRs have decreased Brownian diffusion coefficients compared with IgG1 BCRs (*Figure 3H*). Taken together, these data suggest that IgE BCRs form more prominent clusters with reduced mobility compared with IgG1 BCRs in the absence of cognate antigen, providing further evidence that these BCRs differ in their constitutive activity.

## BCR signaling contributes to antigen-independent PC differentiation

To determine whether BCR signaling was actually mediating the antigen-independent PC differentiation, we tested whether genetic or pharmacological disruption of BCR signaling pathways would affect PC differentiation in the cell culture assay. Treatment of the primary B cell cultures with ibrutinib to inhibit Btk, an important kinase in BCR signaling, led to reduced antigen-independent PC differentiation (*Figure 4A*), while cell viability and CSR were maintained (data not shown). Inhibitors of Syk and PI3Ks, which are critical kinases in BCR signal transduction, also partially inhibited PC differentiation but led to a marked loss of cell viability and thus were not further evaluated (data not shown). To test the role of Syk by a genetic approach without disrupting B cell development, we generated *Syk* heterozygous B cells in vitro by culturing B cells from mice carrying a single loxP-flanked allele of *Syk* ($Syk^{flox/+}$) (*Saijo et al., 2003*) and Cγ1-Cre, which is induced in B cells cultured with anti-CD40 and IL-4 (*Casola et al., 2006*). *Syk* heterozygosity led to reduced PC differentiation in the absence of antigen (*Figure 4B*). The BCR co-receptor CD19 has been implicated in tonic BCR signaling (*Mattila et al., 2013*), as has one of its major targets PI3K (*Srinivasan et al., 2009*). Strikingly, antigen-independent PC differentiation was completely abrogated in CD19-deficient B cells (*Figure 4C*). In contrast, the BCR signaling adapter BLNK (BASH, SLP-65) only partially contributed to antigen-independent PC differentiation, with a two-fold reduction observed in BLNK-deficient B cells (*Figure 4D*). These results suggest that antigen-independent PC differentiation has a differential reliance on particular BCR signaling pathways. Taken together, these data in general demonstrate that BCR signaling is needed for antigen-independent PC differentiation, providing further evidence that this is mediated by constitutive activity of the IgE BCR.

## The IgE BCR constitutive activity is weaker than antigen-dependent signaling

To further evaluate the constitutive activity of the IgE BCR, we compared the effects of perturbing BCR signaling on antigen-independent versus antigen-dependent PC differentiation. With our retroviral transduction system described above, we ectopically expressed TNP-specific light chains together with TNP-specific heavy chains coupled to IgE versus IgG1 constant regions (with the construct shown in *Figure 1E*). We then treated cells with ibrutinib in order to inhibit Btk, prior to antigen stimulation with TNP-OVA. In the absence of TNP-OVA, ibrutinib treatment reduced antigen-independent PC differentiation mediated by the transduced BCRs, as we had previously observed in

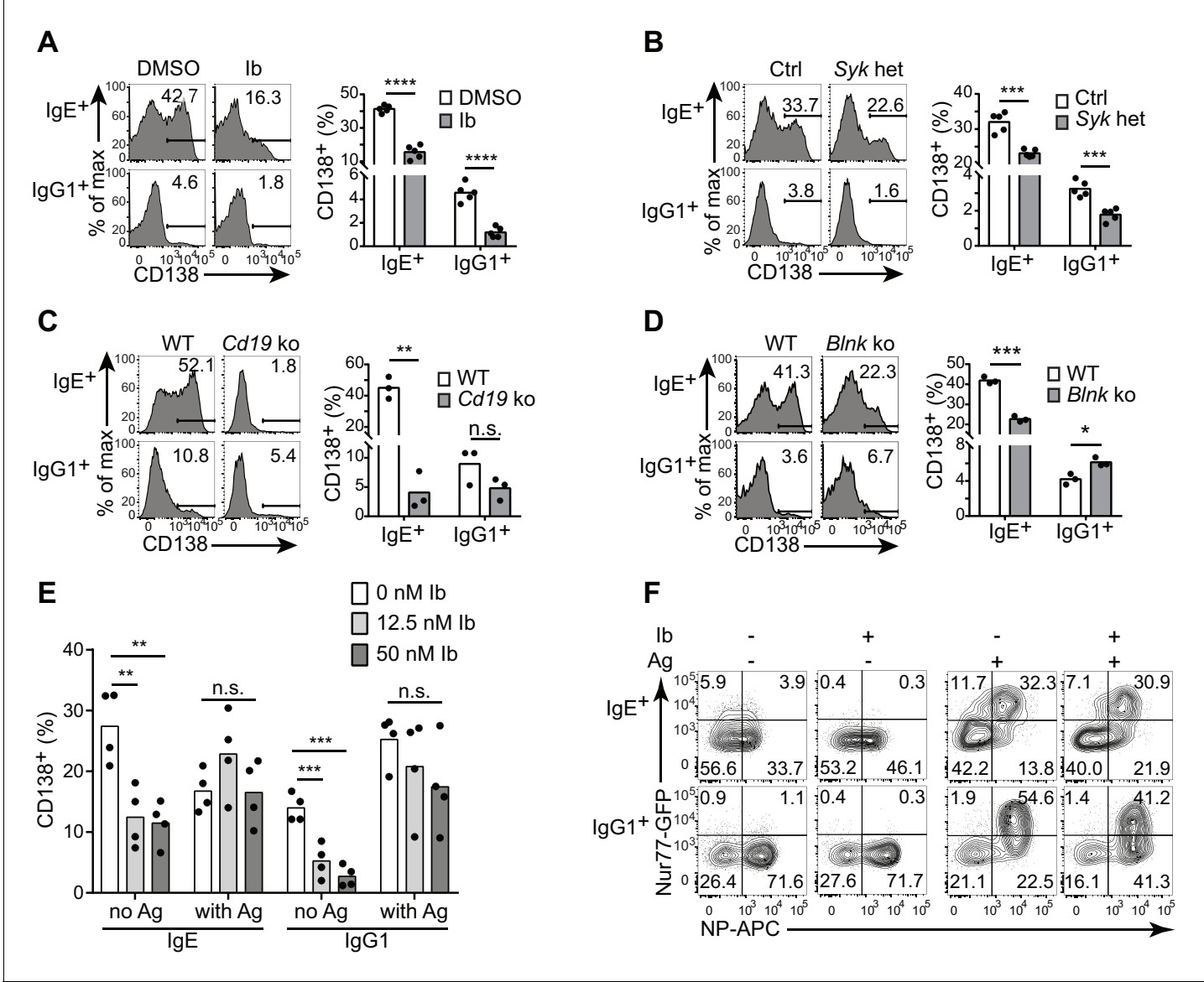

**Figure 4.** Antigen-independent PC differentiation mediated by the IgE BCR is sensitive to perturbations in BCR signaling. B cells were cultured with IL-4 and anti-CD40 for 4 d. (A–D) Representative flow cytometry (left) and quantification (right) of PC differentiation (CD138+) among B cells that were treated with DMSO solvent control versus 12.5 nM ibrutinib (Ib) (A), from control (Ctrl) *Syk*+/+ Cγ1Cre/+ versus *Syk*flox/+ Cγ1Cre/+ (*Syk* het) mice (B), from wild-type (WT) control versus *Cd19*Cre/Cre (*Cd19* ko) mice (C), or from wild-type (WT) control versus *Blnk*−/− (*Blnk* ko) mice (D). Cells were gated as in *Figure 1A*. See also *Figure 4—figure supplement 1*. (E) Quantification of the frequency of PCs (CD138+) among B1-8flox/+ Cγ1Cre/+ B cells retrovirally transduced with TNP-specific IgE or IgG1 BCRs. Ibrutinib (Ib) was added immediately after spinfection (d 1), antigen (TNP-OVA) was added on d 2, and cells were analyzed on d 4. Transduced cells were identified as IgM−IgD−Cerulean+. (F) Flow cytometry of GFP expression in B1-8i, Nur77-GFP B cells. 12.5 nM ibrutinib (Ib) was added on d 2 and then the cognate antigen NP-APC (Ag) was added on d 3, and cells were analyzed on d 4 with further staining on ice with NP-APC to detect antigen-specific cells. Data are representative of two experiments. Dots represent data points from individual experiments. Bars represent the mean. *p<0.05, **p<0.01, ***p<0.001, ****p<0.0001 (t-tests with the Holm-Sidak correction for multiple comparisons (A–D), one-way ANOVA followed by Dunnett's post-test (E)).

The following figure supplement is available for figure 4:

**Figure supplement 1.** IRF-4 contributes to antigen-independent PC differentiation mediated by the IgE BCR in cell culture.

normal primary B cells that had undergone natural class switch recombination to IgE and IgG1 (*Figure 4E*). Interestingly, however, when we added TNP-OVA, antigen-dependent PC differentiation was not significantly affected by ibrutinib treatment (*Figure 4E*). To further evaluate the effects of Btk inhibition on constitutive versus antigen-dependent BCR signals, we used the Nur77-GFP reporter to measure BCR signaling activity in B cells carrying the B1-8 Ig heavy chain variable region knock-in specific for 4-hydroxy-3-nitrophenylacetyl (NP) when paired with λ light chains. Consistent with our previous results, in the absence of antigen, IgE$^+$ B cells exhibited higher Nur77-GFP expression than IgG1$^+$ B cells (*Figure 4F*). The addition of cognate antigen (NP-OVA) resulted in much stronger GFP expression in antigen-specific B cells of both isotypes (*Figure 4F*). Ibrutinib treatment abrogated Nur77-GFP expression in the absence of antigen, whereas ibrutinib treatment had less pronounced effects on Nur77-GFP expression in the presence of antigen (*Figure 4F*). Taken together, these data indicate that the constitutive activity of the IgE BCR is weaker than antigen-dependent BCR stimulation and is more sensitive to pharmacological inhibition.

## BCR signaling constrains in vivo IgE$^+$ GC B cell responses

Based on our above findings that the IgE BCR has a weak but constitutive activity that is distinct from the IgG1 BCR, we anticipated that perturbing BCR signaling in vivo might have differential effects on IgE versus IgG1 responses. After immunization, BLNK-deficient mice showed a striking increase in IgE$^+$ B cell frequencies within GCs, compared with no change in IgG1$^+$ B cell frequencies, in the context of relatively normal total GC B cell numbers (*Figure 5A* and *Figure 5—figure supplement 1A*). BLNK-deficient mice also had an increase in IgE$^+$ PCs, but not IgG1$^+$ PCs (*Figure 5A*). This result differed from cell culture, which we consider in depth below (see the Discussion section). While CD19-deficient mice are defective in T-dependent immune responses in vivo (*Rickert et al., 1995*) and could not be studied, we tested the role of CD19 by immunizing mice heterozygous for *Cd19* (*Rickert et al., 1995*). The two-fold reduction in *Cd19* in these mice also resulted in a selective increase in the frequency of IgE$^+$ B cells in GCs, but did not affect the frequency of IgG1$^+$ B cells nor total GC B cell numbers (*Figure 5A* and *Figure 5—figure supplement 1A*). *Cd19* heterozygous mice also showed a modest elevation in IgE$^+$ PCs (*Figure 5A*). Since both BLNK and CD19 are involved in B cell development, which could potentially have an indirect effect on IgE responses, we also tested the effects of transient pharmacological inhibition of BCR signaling. Wild-type mice were immunized and then 6 d later, at the beginning of the GC and PC responses, were treated with ibrutinib to inhibit Btk for 3 d. This inhibition of BCR signaling during an ongoing GC response resulted in a general two-fold reduction in GC cell numbers (*Figure 5—figure supplement 1B*) as previously reported (*Mueller et al., 2015*), but there was again a selective increase in the frequency of IgE$^+$ B cells in GCs, whereas the frequency of IgG1$^+$ GC B cells was unaffected (*Figure 5B*). A trend toward an increase in IgE$^+$ PCs was also observed in ibrutinib-treated mice, although this did not reach statistical significance (*Figure 5B*). Since ibrutinib can also inhibit Itk (*Dubovsky et al., 2013*), which has been implicated in T cell regulation of IgE responses (*Felices et al., 2009*), we sought to test the effects of BCR signaling perturbation specifically in activated B cells in mice with normal B cell development. In *Syk*$^{flox/+}$ Cγ1$^{Cre/+}$ mice, two functional copies of the *Syk* gene are present in all cells except activated B cells, which express Cγ1-Cre and become *Syk* heterozygous. This two-fold decrease in *Syk* specifically in activated B cells also resulted in a selective increase in the frequency of IgE$^+$ B cells within GCs, whereas IgG1$^+$ B cell frequencies were not significantly affected, in the context of slightly elevated total GC B cell numbers (*Figure 5C* and *Figure 5—figure supplement 1C*). IgE$^+$ PCs, but not IgG1$^+$ PCs, were elevated as well when activated B cells were *Syk* heterozygous (*Figure 5C*). Thus, with four different BCR signaling perturbations, IgE$^+$ B cell frequencies were selectively elevated in GCs, indicating that in general, BCR signaling negatively regulates IgE$^+$ GC B cell responses. The effects of these signaling perturbations on IgE$^+$ PC numbers were variable, possibly reflecting differential requirements for these signaling adapters in antigen-independent versus antigen-dependent PC differentiation (see the Discussion section).

One of the targets of antigen receptor signaling is the transcription factor IRF4, which has been implicated in both GC and PC responses depending on its abundance and association with other transcription factors (*Nutt et al., 2011*; *Ochiai et al., 2013*). While IRF4 is necessary for CSR to IgE and IgG1, precluding the analysis of *Irf4*-deficient B cells, we were able to evaluate *Irf4*-heterozygous B cells, which underwent relatively normal CSR. A single copy of *Irf4* was conditionally deleted in activated B cells from *Irf4*$^{flox/+}$ Cγ1$^{Cre/+}$ mice. Cell culture of *Irf4*-heterozygous B cells reduced the

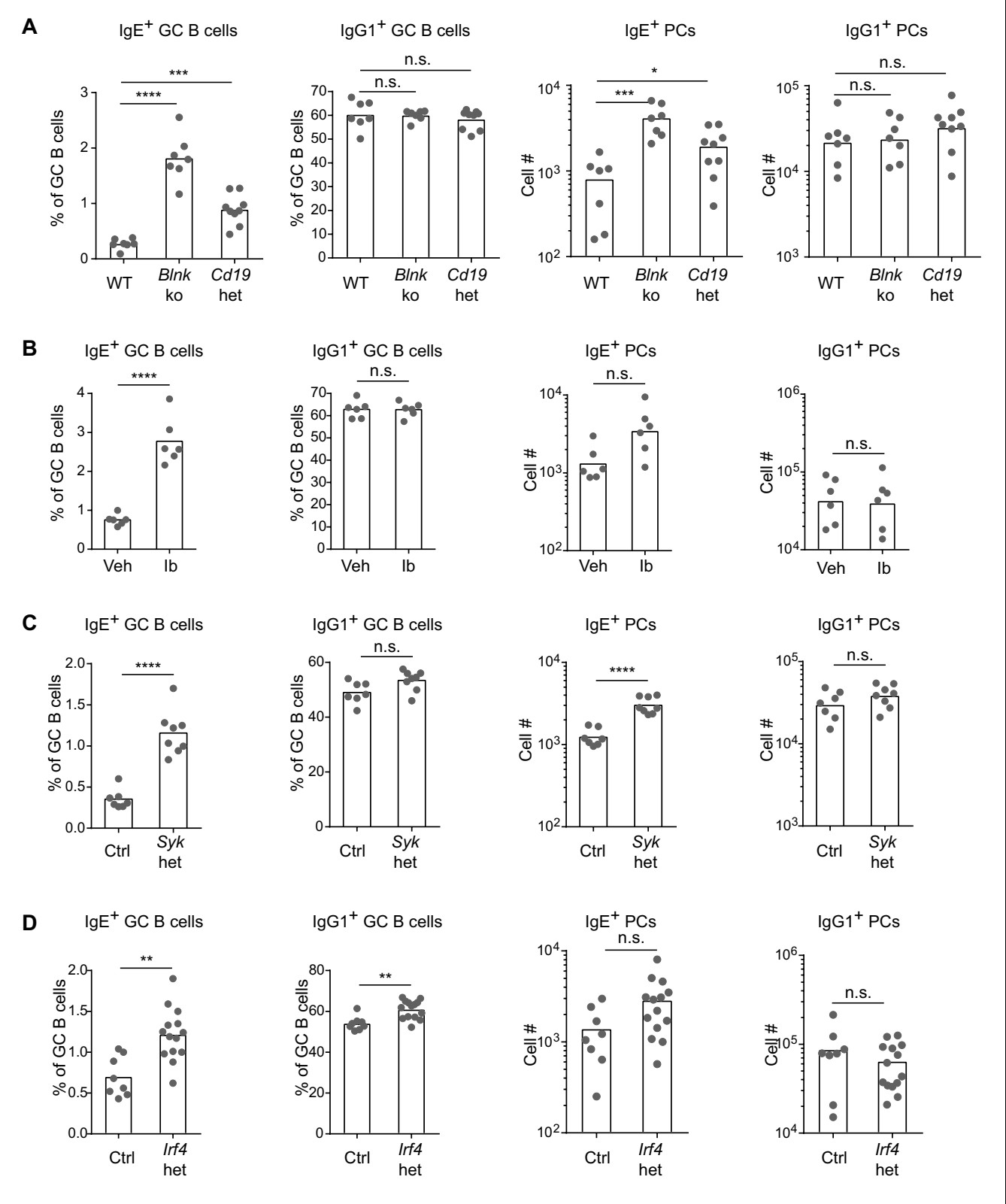

**Figure 5.** BCR signaling negatively regulates in vivo IgE[+] responses. Mice were immunized subcutaneously with NP-CGG in alum adjuvant and draining lymph nodes were analyzed by flow cytometry 9 d later. (A to D) Quantification of frequency of IgE[+] and IgG1[+] cells among GC B cells and the total number of IgE[+] and IgG1[+] PCs in wild-type (WT) versus *Blnk*[−/−] (*Blnk* ko) versus *Cd19*[Cre/+] (*Cd19* het) mice (A), Captex 355 vehicle (Veh) versus ibrutinib (Ib)-treated mice (B), control *Syk*[+/+] Cγ1[Cre/+] (Ctrl) versus *Syk*[flox/+] Cγ1[Cre/+] (*Syk* het) mice (C), control *Irf4*[+/+] Cγ1[Cre/+] (Ctrl) versus *Irf4*[flox/+] Cγ1[Cre/+] (*Irf4*

*Figure 5 continued on next page*

*Figure 5 continued*

het) mice (D). The total number of lymph node cells and GC B cells are provided in *Figure 5—figure supplement 1*. An analysis of somatic hypermutation related to (C) is provided in *Figure 5—figure supplement 2*. PCs were CD138$^+$B220$^{lo-int}$CD38$^{lo}$IgD$^{lo}$ and intracellular IgE$^{hi}$ or total IgG1$^{hi}$. GC B cells were CD138$^-$B220$^{hi}$PNA$^{hi}$CD38$^{lo}$IgD$^{lo}$ and intracellular IgE$^{int}$ or total IgG1$^{int}$. Dots represent individual mice. Bars represent the mean (% of GC B cells) or geometric mean (cell number). Data are representative of two (A), three (C), and four (B) independent experiments. n.s., not significant; **p<0.01; ***p<0.001; ****p<0.0001 (one-way ANOVA followed by Dunnett's post-test (A), t-tests with the Holm-Sidak correction for multiple comparisons (B–D); the numbers of PCs were log transformed for all statistical tests).

The following figure supplements are available for figure 5:

**Figure supplement 1.** Number of total lymph node cells and GC B cells in mice with perturbations in BCR signaling.
**Figure supplement 2.** Frequency of somatic mutations in GC B cells versus PCs.

antigen-independent PC differentiation of IgE$^+$ B cells (*Figure 4—figure supplement 1*), similar to our BCR signaling perturbations in *Figure 4*. These data confirm a role for IRF4 in antigen-independent PC differentiation mediated by the IgE BCR. In vivo, the effects of *Irf4* heterozygosity were subtle, but we observed a two-fold increase in the frequency of IgE$^+$ B cells compared with a 10% increase in the frequency of IgG1$^+$ B cells in GCs (*Figure 5D*). There was also a trend toward increased IgE$^+$ PCs when *Irf4* was made heterozygous, although this did not reach statistical significance (*Figure 5D*). The selective increase in IgE$^+$ B cells in GCs and slight increase in IgE$^+$ PCs when *Irf4* was heterozygous resembled our above findings with perturbations in BCR signaling. These findings suggest that *Irf4* may be one target responsible for the specific effects of BCR signaling on IgE GC and PC responses.

In general, our observations with the various perturbations above showed that changes in IgE$^+$ GC B responses were not always coupled with equivalent changes in IgE$^+$ PC responses, which could be due to heterogeneity in PC responses. Indeed, IgE$^+$ PCs can arise both via extrafollicular and GC-derived pathways (*Yang et al., 2014*). We therefore sought to gain insight into the origin of the IgE$^+$ PCs that accumulated in some mice with perturbed BCR signaling. The extent of somatic mutation of the antibody variable regions in PCs and GC B cells was assessed in conditional *Syk* heterozygous and control mice. We sequenced VDJs containing the VH186.2 variable region, which dominates the NP-specific response on the C57BL/6 background (*Bothwell, 1984*), from individual NP-specific cells. The majority of IgE$^+$ and IgG1$^+$ GC B cells carried somatic mutations, as expected, whereas the vast majority of IgE$^+$ and IgG1$^+$ PCs had germline sequences at this timepoint (*Figure 5—figure supplement 2*). This result indicates that the majority of IgE$^+$ PCs were likely derived from the extrafollicular pathway, suggesting that a two-fold reduction in *Syk* led to an increase in the IgE$^+$ PC extrafollicular response. However, there was an increase in the fraction of PCs carrying somatic mutations when the cells were *Syk* heterozygous. In addition, a substantial fraction of GC B cells still had germline sequences at this timepoint. Therefore, we cannot formally exclude that some of the increase in IgE$^+$ PCs may have arisen from the enhanced IgE GC response of *Syk* heterozygous B cells. Overall, our data implicate BCR signaling in the regulation of both extrafollicular and GC pathways of the IgE response. The IgE BCR may therefore have additional influences on in vivo responses distinct from promoting antigen-independent PC differentiation, which we explored further below.

## The IgE BCR does not directly promote intrinsic apoptosis

A recent study concluded that the IgE BCR negatively regulates IgE responses by promoting high levels of apoptosis through a mitochondrial pathway (*Laffleur et al., 2015*). The apoptosis was linked to relocalization of the mitochondrial protein Hax1 (*Laffleur et al., 2015*), which was reported to bind a sequence found in the CT of the IgE BCR (*Oberndorfer et al., 2006*). *Laffleur et al. (2015)* reported that in primary B cell cultures with anti-CD40 and IL-4, upon withdrawal of these stimuli, IgE$^+$ B cells exhibited increased apoptosis compared with IgG1$^+$ B cells. However, we were unable to reproduce this result. While withdrawal of anti-CD40 and/or IL-4 led to a general increase in apoptotic cells as measured by annexinV staining, the frequency of apoptotic IgE$^+$ and IgG1$^+$ B cells were similar (*Figure 6A*). Ectopic expression of the IgE BCR in primary B cells also resulted in

similar frequencies of apoptosis to ectopic expression of the IgG1 BCR (*Figure 6B and C*). Swapping (constructs 2 and 5) or truncating the IgE CT domain to the short KVK sequence found in IgM (construct 3) also had no significant impact on apoptosis (*Figure 6B and C*). *Laffleur et al. (2015)* also reported that transfection of the IgE BCR into the A20 B cell line promoted high levels of apoptosis, making it difficult to obtain stable transfectants. While the A20 cell line is not amenable to retroviral transduction, we were able to use this approach to express our constructs in WEHI-231 cells, which are known to readily undergo apoptosis upon BCR cross-linking (*Benhamou et al., 1990*; *Hasbold and Klaus, 1990*). Similar frequencies of apoptosis were observed in WEHI-231 cells transduced with the IgE BCR versus IgG1 BCR or with CT mutant constructs (*Figure 6D*), as we had observed in primary B cells. In contrast, anti-IgM treatment induced robust apoptosis of WEHI-231 cells, as expected, confirming the validity of the assay (*Figure 6D*, rightmost panels). The frequency of cells transduced with the IgE BCR was also stable from 3 d to 7 d after transduction (*Figure 6D*), arguing against any deleterious effects of expressing the IgE BCR or its CT. Similar results were also obtained in two other B cell lines, BAL 17 and M12 (*Kim et al., 1979*; data not shown). Notably, *Laffleur et al. (2015)* had expressed human IgE heavy chains in a mouse B cell line without verifying that the human heavy chains can pair appropriately with mouse BCR components, whereas in our study we expressed normal mouse IgE heavy chains in mouse B cell lines, avoiding this potential technical issue.

An in vivo study also reported that IgE[+] GC B cells exhibited higher rates of apoptosis than IgG1[+] GC B cells, as revealed with an active caspase antibody or active caspase substrate (*He et al., 2013*). However, when we stained with an active caspase-3 antibody, we found similar frequencies of apoptosis among IgE[+] and IgG1[+] GC B cells (*Figure 6E and F*). We also observed similar rates of apoptosis when we incubated the cells with an active caspase substrate and when we stained the cells with annexin V (*Figure 6G and H*). Taken together, we find no evidence that the IgE BCR directly promotes intrinsic apoptosis in primary B cells in culture, in B cell lines, nor in vivo GC B cells.

We also confirmed by a genetic approach that intrinsic apoptosis was not the major mechanism responsible for the progressive loss of IgE[+] B cells from GCs over time. In transgenic mice overexpressing the anti-apoptotic gene *Bcl2* in B cells, we had previously reported a dramatic increase in IgE[+] PC but not GC B cell responses (*Yang et al., 2012*). We further characterized the kinetics of IgE[+] GC B cell responses in *Bcl2* transgenic mice for four weeks after immunization. IgE[+] B cells were lost from GCs at a similar rate in *Bcl2* transgenic mice and littermate controls (*Figure 6I*). Taken together, our data are inconsistent with the model that IgE[+] B cells are eliminated from GCs primarily due to increased rates of intrinsic apoptosis mediated by the IgE BCR.

## IgE[+] GC B cells have prolonged cell cycles

Our findings that the IgE BCR does not directly promote apoptosis suggested that other mechanisms are likely responsible for the selective effects of BCR signaling on IgE[+] GC B cell responses. Recent studies have demonstrated that selection for high affinity B cells in the GC is linked to differential proliferation rates (*Gitlin et al., 2015*, *2014*). Cells with faster cell cycles would outcompete cells with slower cell cycles, which can be measured by administering a pulse of a thymidine analog during S phase and then evaluating cell cycle progression (*Gitlin et al., 2015*). Using this approach, we immunized mice and 6–7 d later, at the peak of the GC response, we injected the thymidine analog EdU to pulse-label cells in S phase. We then waited 3.5 hr to allow cells to variably progress from S phase to the G2, M, and/or G1 phases of the cell cycle, which was assessed by measuring DNA content (*Figure 7A*). In wild-type mice, we observed that IgE[+] GC B cells tended to have slower cell cycles than IgG1[+] GC B cells, as revealed by a smaller proportion of EdU[+] cells reaching the G1 phase of the cell cycle in this time window (*Figure 7A and B*). We then tested whether inhibition of Btk with ibrutinib affected the cell cycle speed. Interestingly, ibrutinib treatment resulted in a greater fraction of EdU-labeled cells progressing to the G1 phase of the cell cycle in 3.5 hr, indicating faster cell cycles (*Figure 7C*). In ibrutinib-treated mice, IgE[+] and IgG1[+] GC B cells showed equivalent cell cycle speeds (*Figure 7C*). Thus, inhibiting BCR signaling actually made IgE[+] GC B cells more competitive with IgG1[+] GC B cells. The observed differences in cell cycle speeds are likely to magnify over multiple cell cycles, as GC B cells have been reported to undergo cell division every 6–12 hr (*Allen et al., 2007b*; *Hauser et al., 2007*; *MacLennan, 1994*). These data suggest that BCR signaling normally constrains IgE[+] GC B cell responses through prolonged cell cycle times.

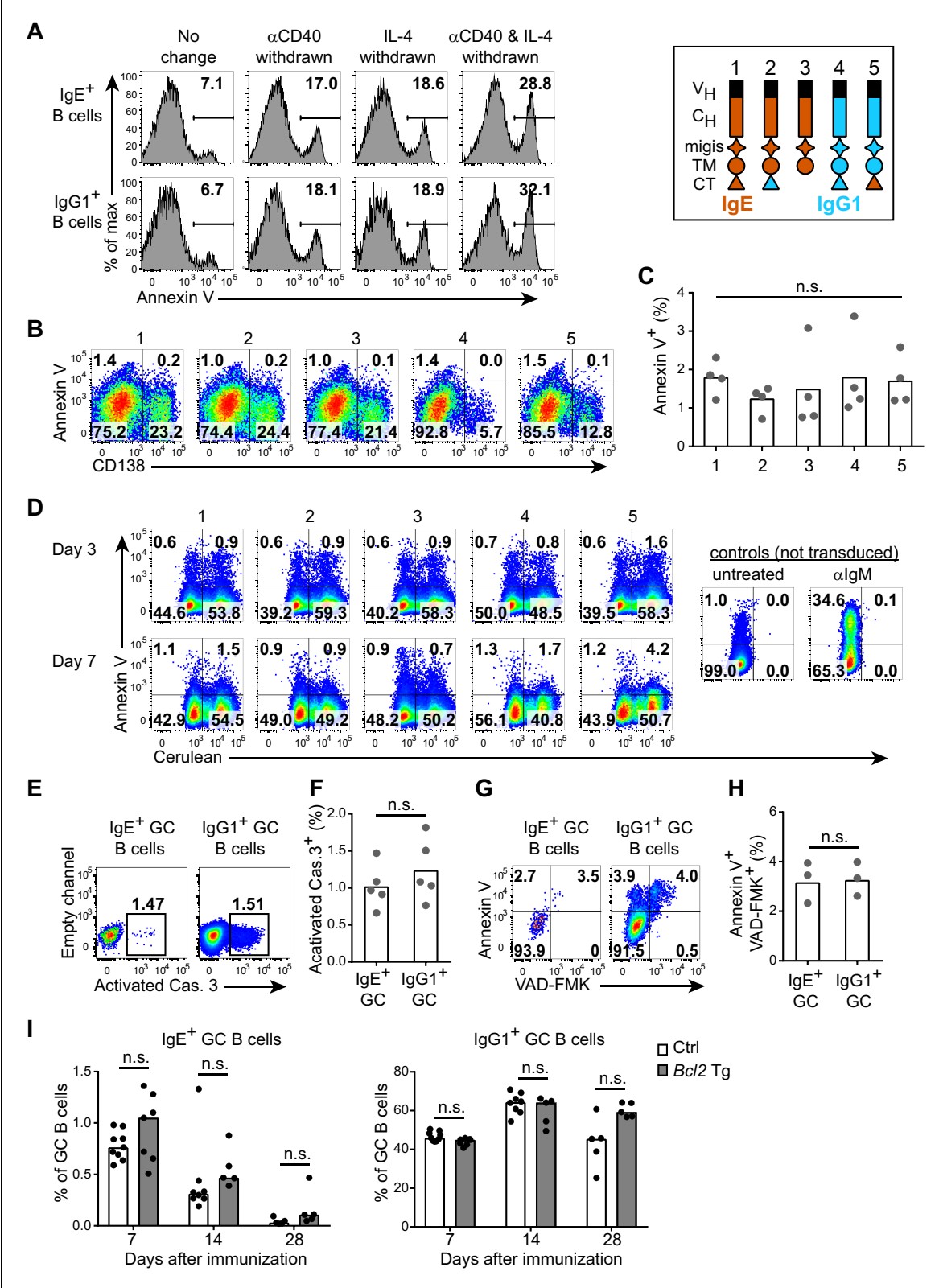

**Figure 6.** The IgE BCR does not promote intrinsic apoptosis in vitro or in vivo. (**A**) Representative flow cytometry of the frequency of apoptosis (annexin V$^+$) of wild-type B cells cultured for 4 d with IL-4 and anti-CD40 (αCD40). The culture media was left unchanged, or replaced with fresh media without anti-CD40 (αCD40 withdrawn), without IL-4 (IL-4 withdrawn), or without both anti-CD40 and IL-4 (αCD40 and IL-4 withdrawn) 5 hr before annexin V staining. Cells were gated as B220$^+$IgD$^-$IgM$^-$ CD138$^-$. (**B** and **C**) Representative flow cytometry (**B**) and quantification (**C**) of the frequency of apoptosis
*Figure 6 continued on next page*

*Figure 6 continued*

(annexin V[+]) of B1-8[flox/+] Cγ1[Cre/+] B cells that were retrovirally transduced with the indicated constructs (refer to the legend in the upper-right) on d 1 of culture. Cells were gated as Cerulean[+]. (**D**) Representative flow cytometry of the frequency of apoptosis (annexin V[+]) of WEHI-231 cells retrovirally transduced with the indicated constructs (refer to the legend in the upper-right) compared to negative (untreated) or positive controls (αIgM treated) and then analyzed after the indicated number of days. Data are representative of three independent experiments. (**E–F**) Representative flow cytometry (**E**) and quantification (**F**) of the frequency of apoptosis (positive for an antibody to activated caspase (Cas.) 3) among IgE[+] versus IgG1[+] GC B cells in draining lymph nodes 7 d after immunization. Similar results were observed in three independent experiments. (**G, H**) Representative flow cytometry (**G**) and quantification (**H**) of the frequency of apoptosis ('Casp-Glow' VAD-FMK[+], annexin V[+]) among IgE[+] versus IgG1[+] GC B cells 7 d after immunization. (**I**) Quantification of frequency of IgE[+] and IgG1[+] cells among GC B cells in littermate control (Ctrl) versus Eμ-*Bcl2-22* transgenic (Tg) mice the indicated number of days after immunization. Data were pooled from four experiments. Mice were immunized subcutaneously with NP-CGG (**E–H**) or NP-KLH (**I**) in alum and draining lymph nodes were analyzed by flow cytometry. GC B cells were gated as CD138[−]B220[hi]PNA[hi]CD38[lo]IgD[lo] and intracellular IgE[int] or total IgG1[int]. Dots represent samples from individual experiments (**C**), or mice (**F, H, I**). Bars represent the mean (**C, F, H**) or median (**I**). n.s., not significant (one-way ANOVA (**C**), paired t-tests (**F, H**), and t-tests with the Holm-Sidak correction for multiple comparisons (**I**)).

## Poor antigen uptake and presentation by IgE[+] GC B cells

As previous studies have linked GC B cell proliferation and selection to the acquisition of T cell help (*Gitlin et al., 2015*, *2014*), we hypothesized that the IgE[+] B cells were at a disadvantage for T cell interactions. It is thought that the ability of the membrane BCR to bind and endocytose antigen, which is then processed and presented on MHC class II, determines the extent of T cell interactions in the GC (*Allen et al., 2007a*; *Victora and Nussenzweig, 2012*). The BCR was reported to be in lower abundance on the surface of IgE[+] GC B cells compared with IgG1[+] GC B cells (*He et al., 2013*). In order to confirm this finding, we used an adoptive transfer system in which we tracked antigen-specific B cells. Specifically, Hy10 knock-in B cells (*Allen et al., 2007b*), specific for avian egg lysozyme, were transferred into congenically-marked mice, which were then immunized with the low affinity antigen duck egg lysozyme (DEL), conjugated to OVA to provide T cell help. We then evaluated BCR surface expression by measuring the capacity for antigen binding. Hy10 IgE[+] and IgG1[+] GC B cells were labeled with fluorescent hen egg lysozyme (HEL), a high affinity antigen. We observed that antigen-specific IgE[+] GC B cells bound 4.4-fold less HEL than antigen-specific IgG1[+] GC B cells (*Figure 7D and E*), consistent with a four-fold difference reported with a different antigen system (*He et al., 2013*). We then tested antigen uptake and presentation with the well-established Y-Ae system (*Germain and Jenkins, 2004*), which has recently been applied to discern differences in the abundance of peptide-MHC complexes on the surface of GC B cells (*Bannard et al., 2016*). In this system, the Y-Ae antibody recognizes the Eα peptide presented on the I-A[b] MHC molecule (*Murphy et al., 1989*; *Rudensky et al., 1991*). We conjugated the Eα peptide to HEL and then administered this reagent in the aforementioned adoptive transfer system. After several hours, allowing time for antigen processing to occur, IgE[+] GC B cells had reduced Y-Ae antibody binding compared with IgG1[+] GC B cells, indicating that IgE[+] GC B cells displayed fewer peptide-MHC complexes (*Figure 7F and G*). Taken together, these data suggest that the low BCR surface expression on IgE[+] GC B cells limits the ability of these cells to compete for T cell help, illuminating an additional mechanism by which the IgE BCR may limit IgE[+] GC B cell responses.

## Discussion

The findings we have reported here establish that distinct properties of the IgE BCR regulate the fate of IgE[+] B cells. In the presence of stimuli mimicking T cell help, the constitutive activity of the IgE BCR promoted PC differentiation in an antigen-independent manner. In contrast, the IgG1 BCR promoted PC differentiation only when the receptor was ligated with cognate antigen. This difference between the IgE and IgG1 BCRs was attributed to multiple domains, particularly the extracellular membrane-proximal migis region, with further contributions from the IgE CH2 and CH3 extracellular domains and the IgE CT. In cell culture, the BCR signaling adapters BLNK, Btk, CD19, and Syk contributed to antigen-independent PC differentiation mediated by the IgE BCR, with an essential role for CD19. In mice, genetic deficiency or pharmacological inhibition of these signaling components consistently resulted in elevated IgE GC responses, yet IgE PC responses were variably affected. The increased IgE[+] GC B cell responses upon Btk inhibition were associated with

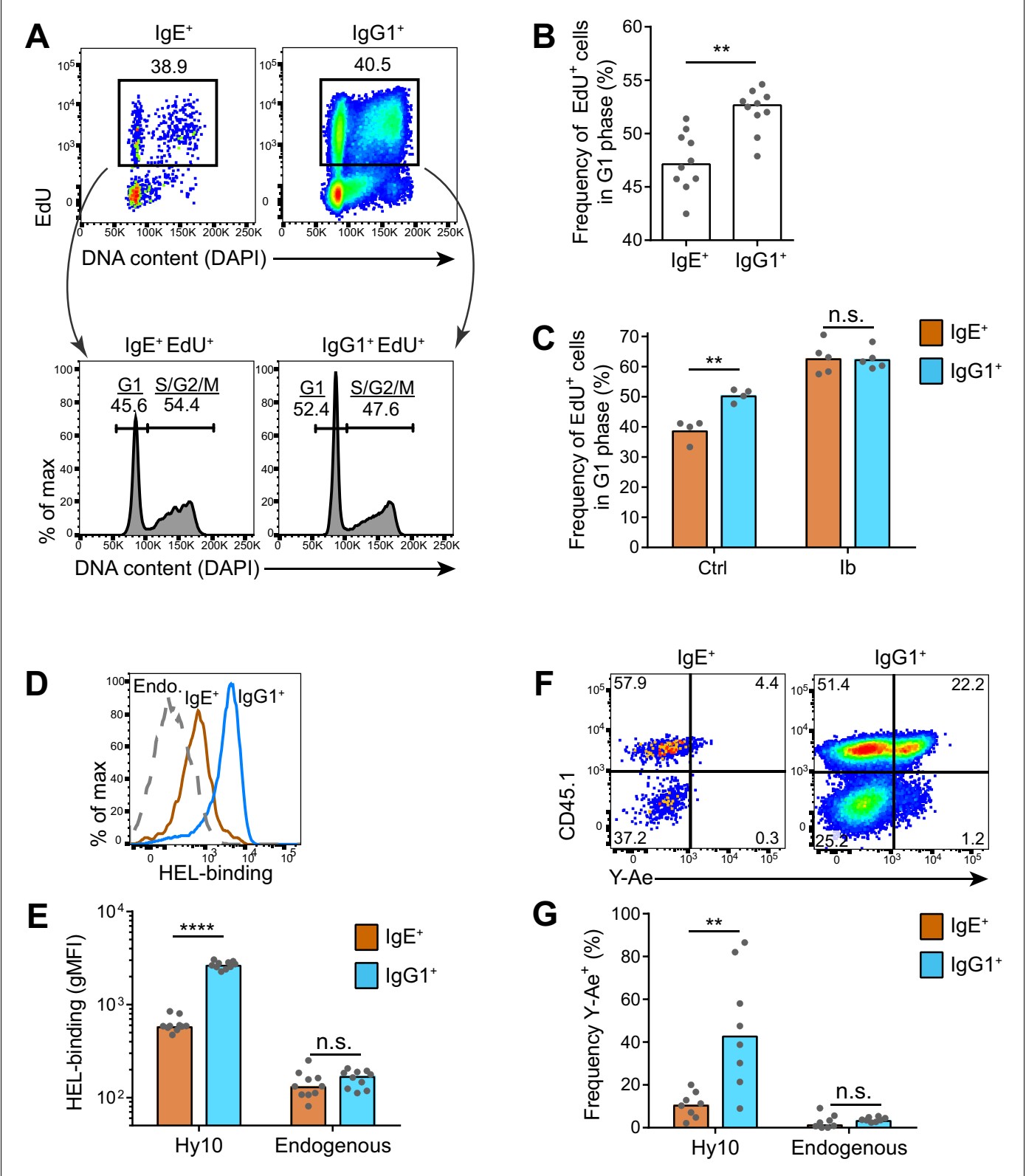

**Figure 7.** IgE+ GC B cells exhibit delayed cell cycle progression and reduced antigen presentation. (A–C) Mice were injected with EdU to pulse-label cells in S phase 6.5 d after subcutaneous immunization with NP-CGG in alum. Draining lymph nodes were harvested 3.5 hr later. (A) Representative flow cytometry of EdU staining and DAPI labeling (DNA content) in IgE+ and IgG1+ GC B cells. EdU+ cells were gated (upper panels) and G1 versus S/G2/M phase cells were resolved by DNA content (lower panels). (B) Quantification of the frequency of EdU+ cells in G1 phase among IgE+ versus IgG1+ GC B

*Figure 7 continued on next page*

*Figure 7 continued*

cells. (**C**) Quantification of the frequency of EdU$^+$ cells in G1 phase among IgE$^+$ versus IgG1$^+$ GC B cells from mice treated with vehicle (Ctrl) or ibrutinib (Ib) twice daily for 3 d, starting 4 d after immunization. (**D–G**) HEL-specific (Hy10) B cells were transferred into wild-type congenic recipient mice and then mice were immunized subcutaneously with DEL-OVA in alum and draining lymph nodes were analyzed 7 d later. (**D** and **E**) Representative flow cytometry (**D**) and quantification (**E**) of BCR surface expression by HEL-Alexa647 labeling of Hy10 GC B cells (CD45.2$^+$) compared with recipient endogenous (Endo.) GC B cells (CD45.1$^+$). (**F** and **G**) Representative flow cytometry (**F**) and quantification (**G**) of Eα peptide-MHC II antigen presentation by Y-Ae antibody staining of Hy10 GC B cells (CD45.1$^+$) compared with recipient endogenous GC B cells (CD45.1$^-$). GC B cells were gated as CD138$^-$B220$^{hi}$PNA$^{hi}$CD38$^{lo}$IgD$^{lo}$ and intracellular IgE$^{int}$ or total IgG1$^{int}$ in all panels, with the addition of IgM$^{lo}$ (**F** and **G**). Similar results were obtained in two independent experiments. Dots represent individual mice and bars represent the median (**B, E, G**) or mean (**C**). gMFI, geometric mean fluorescence intensity; n.s., not significant; **p<0.01; ****p<0.0001 (Wilcoxon matched-pairs signed rank test (**B**), and t-tests with the Holm-Sidak correction for multiple comparisons (**C, E, G**)).

accelerated cell cycle progression, suggesting that the reduced competitive fitness of IgE$^+$ GC B cells is in part due to BCR signaling. In addition, low surface expression of the IgE BCR on GC B cells led to reduced antigen uptake and presentation, likely limiting access to T cell help.

While previous studies have largely focused on isotype-dependent BCR responses to antigen stimulation, here we revealed that BCR isotypes also differ in antigen-independent activity. BCRs were known to exhibit tonic signaling, as deletion of the BCR results in rapid B cell death (*Kraus et al., 2004*). In this study, we reported that the IgE BCR stood out among all isotypes tested as exhibiting the highest capability of inducing antigen-independent PC differentiation. However, our data suggest that the constitutive activity of the IgE BCR is substantially weaker in magnitude than acute antigen stimulation. Interestingly, weak, constitutive BCR activity also promoted the PC differentiation of B cells from mice in which the BCR had been substituted with the latent membrane protein 2A (LMP2A) from Epstein Barr virus (*Lechouane et al., 2013*). Increased PC differentiation was also reported in a BCR transgenic line with low surface abundance of IgG2a and elevated constitutive activity (*Man et al., 2010*). We therefore propose that weak, constitutive BCR activity may be highly effective at promoting PC differentiation.

Our data implicated the extracellular membrane-proximal migis region as a key difference between the IgE and IgG1 BCRs. The migis was also a major determinant of BCR surface localization in the presence versus the absence of Igα. This finding suggests that the interaction between the BCR and Igα extends beyond the TM domains into the extracellular membrane proximal region. A possible association of the migis region with Igα had been speculated based on experiments with human versus mouse Igα (*Reth, 1992*). In addition, the migis regions of IgM and IgD were reported to control the glycosylation pattern of Igα (*Pogue and Goodnow, 1994*). Since our data identified the migis region as an important component of the antigen-independent activity of the IgE BCR, we propose that this region is normally involved in signal transduction upon antigen binding. The migis of IgE had been implicated in signal transduction in a study of simplified BCR molecules in which receptors were cross-linked (*Poggianella et al., 2006*). Human IgE BCRs contain one of two possible migis regions by alternative splicing, a long versus short form (*Poggianella et al., 2006*). It will be interesting in future studies to examine whether these splice variants differentially regulate the interaction of the IgE BCR with Igα.

Our finding that the CT region of IgE could contribute to, but was not essential for, antigen-independent PC differentiation, is consistent with a role for this region in modulating or enhancing IgE BCR signaling (*Engels et al., 2009*; *Sato et al., 2007*). Some functions of the IgE CT might be equivalent to the IgG CT due to the conserved tyrosine motif; thus, our domain swap experiments would highlight isotype-specific functions of these domains (*Engels et al., 2009*). A contribution of the IgE CT to in vivo responses was noted in a mouse strain engineered to delete this region, however, notably, the effect was quantitative (2–4 fold) rather than absolute (*Achatz et al., 1997*), again suggesting the IgE CT contributes to but is not essential for the distinct features of the IgE BCR.

The domain swap experiments also revealed a partial contribution of the IgE extracellular CH2 and CH3 domains to antigen-independent PC differentiation. Structural studies of secreted IgE indicated that an asymmetric bend can occur in IgE at the CH2-CH3 linker region, causing one of the two CH2 domains to fold back onto the CH3 and CH4 domains, making particularly extensive contacts with the CH3 domain (*Wan et al., 2002*). Recent evidence suggests that this bent shape can

also be extended, which can subsequently allow CH2 to flip from one side of CH3-CH4 to the other (*Drinkwater et al., 2014*). It seems plausible that for the membrane IgE BCR, the ability to adopt an asymmetric bend conformation mimics antigen binding. The IgE CH2 domain versus IgG hinge regions are also analogous to the domain structures of IgM versus IgD, respectively, which recently have been implicated in differential antigen responsiveness (*Übelhart et al., 2015*). Providing further evidence for the regulation of in vivo B cell responses by the IgE extracellular domains, a recent study found that memory responses were abrogated when the extracellular domains of IgG1 were replaced with those of IgE (*Turqueti-Neves et al., 2015*).

Of the BCR signaling adapter molecules that we examined, we observed that CD19 was essential for antigen-independent PC differentiation. Signal transduction via PI3K, a target of CD19, has also been identified as the major pathway by which tonic BCR signaling maintains BCR survival (*Srinivasan et al., 2009*). PI3K may therefore be a general transducer of constitutive BCR signals which can potentially mediate different outcomes including PC differentiation and survival. Since PI3K signaling is also downstream of numerous other receptors, future studies may provide further insights into how these signals are integrated to determine cell fate. Our studies also support a role for Syk, Btk, and BLNK in antigen-independent PC differentiation. In future studies, it would be interesting to delineate how these findings relate to studies on the role of ERK1/2 in antigen-dependent PC differentiation (*Yasuda et al., 2011*).

In physiologic immune responses to T-dependent antigens, B cells may receive both antigen-independent and antigen-dependent BCR signals. Since the IgE BCR exhibited antigen-independent activity that was distinct from the IgG1 BCR and that was particularly sensitive to perturbations in BCR signaling, we hypothesized that modulating BCR signaling would have a selective effect on IgE$^+$ B cell responses in vivo. Indeed, we found it striking that in four different perturbations of BCR signaling, the frequency of IgE$^+$ B cells within GCs was selectively increased, whereas the frequency of IgG1$^+$ B cells was not affected. Together with our above findings, these results suggest that the antigen-independent activity of the IgE BCR normally limits the magnitude of IgE$^+$ B cell responses in the GC.

Unexpectedly, some of the perturbations in BCR signaling that we tested resulted in increased numbers of IgE$^+$ PCs in immunized mice, whereas we had observed reductions in IgE$^+$ PCs in cell culture. This result could be related to the timing of PC differentiation driven by antigen-independent versus antigen-dependent BCR signals. Specifically, the antigen-independent activity of the IgE BCR is likely to promote the 'premature' terminal differentiation of IgE$^+$ B cells into PCs. The BCR signaling perturbations may have inhibited this premature terminal differentiation, thereby enabling IgE$^+$ B cells to undergo further rounds of proliferation, prior to normal antigen-dependent PC differentiation. This model is consistent with our observations that the BCR signaling perturbations did not affect IgG1$^+$ PC numbers, which our data suggest are generated primarily by antigen-dependent signals. In addition, this model is consistent with our findings regarding IRF4, a transcription factor that is essential for PC differentiation and can be induced by BCR signaling (*Nutt et al., 2011*; *Ochiai et al., 2013*). When activated B cells were made *Irf4* heterozygous, this resulted in reduced antigen-independent IgE$^+$ PC differentiation in cell culture but a slight increase, rather than a decrease, in the total number of IgE$^+$ PCs in immunized mice. Some of the increase in IgE$^+$ PCs that we observed with various perturbations in mice could be a consequence of enhanced IgE$^+$ GC B cell responses, but our somatic hypermutation analysis suggests this cannot fully explain the results, since most IgE$^+$ PCs had germline sequences whereas most IgE$^+$ GC B cells had somatic mutations. Another possibility is that the BCR signaling perturbations could have a direct impact on the proliferation or longevity of the IgE$^+$ PCs, which have high surface expression of the IgE BCR. The BCR signaling perturbations could also have affected the rate of CSR to IgE, as PI3K signaling has been reported to negatively regulate CSR (*Omori et al., 2006*). Overall, our data are most consistent with the model that perturbations in BCR signaling led to delayed terminal differentiation of IgE$^+$ B cells, enabling further rounds of proliferation and the generation of longer-lived PCs.

Since the perturbations in BCR signaling led to a very consistent increase in IgE$^+$ GC B cells, whereas the effects on IgE$^+$ PC numbers were less consistent, it seemed likely that IgE BCR signaling does not solely affect PC differentiation. Our mutational analysis revealed that most of the IgE$^+$ PCs were derived from the extrafollicular pathway at the timepoint in which we could observe increases in IgE$^+$ GC B cells, suggesting that BCR signaling also has a direct impact on the GC response. One group proposed that IgE$^+$ B cells are progressively lost from GCs due to low BCR surface

expression, leading to reduced BCR signaling and apoptosis (*He et al., 2013*). However, using similar methods to detect apoptotic B cells, we were unable to replicate the finding that IgE[+] B cells had higher rates of apoptosis compared with IgG1[+] B cells in GCs. A caveat to these disparate findings is that the ex vivo analysis of the apoptosis of GC B cells may be hampered by the rapid uptake of apoptotic cell fragments by tingible body macrophages (*MacLennan, 1994*). In contrast, the analysis of apoptosis in cultured B cells should be more robust. Yet, in cell culture studies, we were also unable to observe an increased rate of apoptosis in IgE[+] B cells compared with IgG1[+] B cells, in contrast to a recent study (*Laffleur et al., 2015*). We further verified that the ectopic expression of the IgE BCR in primary B cells did not promote apoptosis in cell culture. In addition, while a recent study reported that IgE BCR expression promoted the apoptosis of a B cell line (*Laffleur et al., 2015*), we were unable to replicate this result in three different B cell lines. Consistent with our observations, a study of isotype-specific BCR signaling reported no difficulty in expressing the IgE BCR in B cell lines (*Sato et al., 2007*). A possible explanation for the apoptosis due to the IgE BCR observed in the *Laffleur et al. (2015)* study was that the human IgE BCR was expressed in mouse B cells. The human IgE heavy chain may not appropriately pair with key accessory proteins in mouse cells, such as the mouse Ig light chains or mouse Igα, which could affect BCR surface localization. We also observed that the progressive loss of IgE[+] B cells from GCs could not be prevented by overexpression of the anti-apoptotic gene *Bcl2*, in contrast to our prior results demonstrating that short-lived IgE[+] PCs were rescued from apoptosis. Similarly, another group reported that IgE[+] GC B cells were very rare in mice deficient in the death receptor Fas, despite exaggerated IgE[+] PC responses (*Butt et al., 2015*). Taken together, our data are inconsistent with the notion that the IgE BCR directly promotes apoptosis in B cells.

We therefore considered other mechanisms by which IgE[+] GC B cell responses might be restricted by the IgE BCR. Recent studies of the GC have revealed that differential rates of proliferation may contribute to the selection of high affinity B cells (*Gitlin et al., 2015*, *2014*). On average, IgE[+] GC B cells exhibited slower cell cycles than IgG1[+] GC B cells, yet these rates increased and became equivalent when BCR signaling was inhibited. Interestingly, in a recent study, most GC B cells were observed to have dampened BCR signaling due to high expression of the phosphatase SHP-1, yet an exception was the cells in the G2/M phases (*Khalil et al., 2012*). We favor a model in which constitutive signaling by the IgE BCR delays the progression of IgE[+] GC B cells through the G2/M phases of the cell cycle. After multiple cell cycles, this delayed proliferation may result in the IgE[+] GC B cells being outcompeted by IgG1[+] GC B cells.

The low abundance of the IgE BCR on GC B cells, reported in a recent study (*He et al., 2013*) and confirmed here, is likely to also influence GC B cell responses. In addition to an unusual polyadenylation sequence (*Anand et al., 1997*; *Karnowski et al., 2006*), the low expression of the IgE BCR may be related to its constitutive activity, as has been observed with the downregulation of IgM on autoreactive B cells that receive chronic activation signals (*Goodnow et al., 1988*). Indeed, a recent study obtained evidence for spontaneous internalization of the IgE BCR (*Laffleur et al., 2015*), which is likely related to its constitutive activity. Since we could not observe increased apoptosis of IgE[+] GC B cells, we considered the impact of low BCR expression on antigen uptake and presentation. We observed that a smaller proportion of IgE[+] GC B cells captured antigen and displayed peptide-MHC complexes compared with IgG1[+] GC B cells. This finding suggests that IgE[+] GC B cells have limited access to T cell help compared with IgG1[+] GC B cells, due to low expression of the IgE BCR.

Overall, our findings suggest that expression of the IgE BCR may result in a competitive disadvantage for IgE[+] B cells in the GC, due to premature differentiation, delayed cell cycle progression, and limited antigen uptake and presentation. These findings may thus help explain the progressive loss of IgE[+] GC B cells over time observed by multiple groups (*He et al., 2013*; *Talay et al., 2012b*; *Yang et al., 2012*). However, we note that our observations were at a population level and do not directly reflect the behavior of individual cells. A very small fraction of IgE[+] GC B cells could still acquire the high affinity mutations necessary to surpass the stringent threshold of selection during affinity maturation, as has been observed by two groups (*He et al., 2013*; *Yang et al., 2012*). Our findings may also have implications for the dynamics of IgE[+] GC B cells. For example, signals provided by T cell help are thought to promote the transit of GC B cells from the light zone to the dark zone (*Victora and Mesin, 2014*). IgE[+] B cells, at a disadvantage for T cell help, might thus show reduced evidence of light zone to dark zone transit. However, the net impact on positioning within the GC may be hard to predict. Indeed, the signals that regulate the timing at which cells transit

from the dark zone to the light zone are not well defined (*Bannard et al., 2013*; *Victora and Mesin, 2014*). IgE$^+$ GC B cells also exhibited different cell cycle characteristics than IgG1$^+$ GC B cells, which might influence dark and light zone positioning (*Victora and Nussenzweig, 2012*). One study reported a paucity of IgE$^+$ B cells in the GC light zone (*He et al., 2013*), whereas we had readily observed IgE$^+$ GC B cells within the dense follicular dendritic cell network corresponding to the light zone (*Yang et al., 2012*). It would be interesting in future studies to determine whether and how the IgE BCR influences the positioning of cells in dark and light zones over the course of an immune response.

While our manuscript was being finalized, another study was published (*Haniuda et al., 2016*) reporting that autonomous signaling of the IgE BCR induced PC differentiation and apoptosis. In this study, ectopic expression of the IgE BCR promoted PC differentiation in cultured B cells, similar to our findings. This group also reported that the IgE BCR promoted apoptosis; however, we were not able to observe this effect in any of our in vitro or in vivo assays. As in our domain swap studies, this study identified the IgE migis region as critical for antigen-independent PC differentiation. The IgE migis was reported by Haniuda et al. to influence the BCR association with CD19, which we propose would be secondary to the interaction with Igα that we identified here. The extracellular domains of IgE were also implicated in the autonomous activity of the IgE BCR in this paper, which here we have further defined as specifically the CH2 and CH3 domains of IgE. The authors also reported a modest impact of deleting the IgE CT but did not test swapping this domain with that of IgG1, as we reported here. Another distinct feature of our work presented here is the microscopy studies of the mobility and clustering of the IgE BCR, which were not examined by Haniuda et al. While both groups observed increased IgE$^+$ GC B cells in *Cd19*-heterozygous and BLNK-deficient mice, our conclusions differed with respect to IgE$^+$ PC responses. Haniuda et al. reported a decreased IgE PC/GC ratio in *Cd19*-heterozygous mice as evidence of decreased PC generation; however, this ratio was likely affected primarily by the increased number of IgE$^+$ GC B cells, since we actually observed a modest increase, rather than a decrease, in IgE$^+$ PC numbers in these mice. We were also not able to reproduce the decrease in IgE$^+$ PCs reported in BLNK-deficient mice and instead we observed a striking increase in IgE$^+$ PCs. We further tested the in vivo impact of inhibiting Btk as well as deletion of a single copy of *Syk* and *Irf4*, which also led to increased IgE$^+$ GC B cell responses but inconsistent effects on IgE$^+$ PCs. While we support the idea that the autonomous activity of the IgE BCR can promote short-lived PC differentiation, the uncoupling of IgE$^+$ GC B cell and PC responses in our studies suggested that BCR signaling has other impacts on IgE$^+$ GC responses. Specifically, we established that BCR signaling affects the cell cycle progression of IgE$^+$ GC B cells and that low BCR surface expression led to reduced antigen uptake and presentation.

Taken together, we propose the following summary model to account for our findings. Upon CSR to IgE, a B cell will begin to express the IgE BCR, which has distinct constitutive activity. If this B cell receives T cell help, this BCR activity will promote PC differentiation. However, if this B cell does not receive T cell help, this BCR activity will delay cell cycle progression, causing the B cell to be outcompeted by clones expressing other isotypes. In this way, the constitutive activity of the IgE BCR makes cell fate relatively independent of antigen stimulation of the BCR, but rather highly dependent on antigen presentation and T cell help. Thus, the availability of T cell help may be a major mechanism for controlling IgE B cell responses. The chronic activity of the IgE BCR leads to surface BCR down-modulation, reducing antigen uptake and presentation, thus making it less likely that the cell will receive T cell help. Premature PC differentiation, delayed cell cycle progression, and limited T cell help due to IgE BCR expression all result in the progressive decline in IgE$^+$ GC B cells and prevent the generation of IgE$^+$ memory B cells and long-lived PCs, thereby reducing the average affinity and overall duration of the IgE response.

A general implication of our work is that BCR signaling negatively regulates IgE responses. Interestingly, Btk-deficient mice had been observed to undergo enhanced IgE responses, although this was not thought to be due to BCR signaling (*Kawakami et al., 2006*). In genetic experiments that specifically affect signaling in B cells, we were able to demonstrate that diminished BCR signaling leads to exaggerated IgE$^+$ responses. Our data therefore suggest that genetic variations or pharmacological treatments that alter the strength of BCR signaling may have a selective effect on IgE$^+$ responses, which may be clinically important in the development of allergy.

## Materials and methods

### Mice

C57BL/6J mice (RRID:IMSR_JAX:000664), Boy/J CD45.1 mice (RRID:IMSR_JAX:002014; B6.SJL-Ptprc$^a$Pepc$^b$/BoyJ), B1-8i mice (RRID:IMSR_JAX:012642; B6.129P2(C)-Igh$^{tm2Cgn}$/J), Blnk$^{-/-}$ mice (RRID:IMSR_JAX:004524; B6.129-Blnk$^{tm1Achn}$/J), Cγ1-Cre mice (RRID:IMSR_JAX:010611; B6.129P2 (Cg)-Ighg1$^{tm1(cre)Cgn}$/J), Irf4$^{flox}$ mice (RRID:IMSR_JAX:009380; B6.129S1-Irf4$^{tm1Rdf}$/J), and Syk$^{flox}$ mice (RRID:IMSR_JAX:017309; B6.129P2-Syk$^{tm1.2Tara}$/J) were originally from The Jackson Laboratory. Aicda$^{-/-}$ mice (RRID:MGI:2654846; Aicda$^{tm1Hon}$; [Muramatsu et al., 2000]), B1-8$^{flox}$ mice (RRID:MGI: 3693006; Igh$^{tm4Cgn}$; [Lam et al., 1997]), Cd19$^{Cre}$ mice (RRID:MGI:1931143; Cd19$^{tm1(cre)Cgn}$; [Rickert et al., 1995]), Eμ-Bcl2-22 mice (RRID:MGI:3052827; B6.Cg-Tg(BCL2)22Wehi; [Strasser et al., 1991]), Hy10 mice (RRID:MGI:3702732; B6.Cg-Igh$^{tm1Cys/+}$, Tg(Igk)5Cys; [Allen et al., 2007b]), and Nur77-GFP (RRID:MGI:4847273; Tg(Nr4a1-EGFP)GY139Gsat; [Zikherman et al., 2012]) mice were maintained on the C57BL/6 background and/or bred to Boy/J CD45.1 congenic mice. Otherwise, C57BL/6 CD45.1 congenic mice, which were used as wild-type mice in some experiments, were from the National Cancer Institute / Charles River Frederick National Laboratory (01B96; B6-Ly5.2/Cr, later renamed to B6-Ly5.1/Cr). Mice with significant skin lesions or other signs of poor health were excluded from the study. Mice were housed in specific-pathogen-free facilities and protocols were approved by the Institutional Animal Care and Use Committee of the University of California, San Francisco.

### Antigens

NP-CGG (estimated conjugation ratio of 30–33) and NP-KLH (estimated conjugation ratio of 27–29) were purchased from Biosearch Technologies.

For the preparation of DEL-OVA, DEL (Worthington Biochemical) was resuspended at 11 mg/mL in 50 mM sodium phosphate buffer pH 7.1 with 2.2 mM EDTA (Thermo Fisher Scientific), then diluted with 10% v/v 1 M sodium bicarbonate. Endograde OVA (Hygros) was resuspended in PBS at 2 mg/mL. Lysines of DEL were labeled with a free thiol residue by incubating DEL with Traut's reagent (Thermo Fisher Scientific) at an 0.8:1 ratio of Traut's:DEL for 60 min. Concomitantly, a malei-mide group was linked to OVA lysines using sulfosuccinimidyl 4-(N-maleimidomethyl)cyclohexane-1-carboxylate (Sulfo-SMCC; Thermo Fisher Scientific) at a 15:1 ratio of Sulfo-SMCC:OVA. DEL-SH and OVA-maleimide conjugates were respectively purified through Bio-Spin 6 and Bio-Spin 30 columns (Bio-Rad). DEL-SH was added to OVA-maleimide at an estimated 10:1 excess of DEL-SH: OVA-malei-mide based on the original concentrations of reagents. Free DEL was removed from the preparation by gel filtration in DPBS over a Superdex 200 Increase 10/300 GL column (GE Healthcare), followed by concentration through Amicon Ultra 30k MWCO concentrator columns (EMD Millipore). The final concentration was determined from the A$_{280}$ using a 0.1% Cε of 1.18, assuming an average 1:1 stoichiometry of DEL:OVA.

For the preparation of HEL conjugated to Eα peptide, HEL (Sigma-Aldrich) was resuspended at 4 mg/mL in H$_2$O with 2 mM EDTA. A maleimide group was linked to HEL lysines using Sulfo-SMCC at a ratio of 0.7:1 Sulfo-SMCC:HEL for 60 min at room temperature with rolling and tilting. Free Sulfo-SMCC was removed by passing the HEL Sulfo-SMCC mixture through either 7 MWCO Zeba-Spin (Thermo Fisher Scientific) or Bio-Spin 6 (Bio-Rad) columns equilibrated with 50 mM sodium phosphate buffer pH 7.1 with 2 mM EDTA. The resulting HEL-maleimide conjugate was incubated with the Eα peptide carrying an additional terminal cysteine residue (H$_2$N-ASFEAQGALANIAVDKAC-OH; New England Peptide) at a 2:1 ratio of Eα$_{cys}$:HEL, based on the original concentrations. Stoichiometric (1:1) HEL-Eα was purified from the soluble fraction over a HiTrap SP FF cation exchange column (GE Healthcare) on an ÄKTApure chromatography system, using 50 mM sodium phosphate buffer pH 7.1 (solution A) with a gradient into 50 mM sodium phosphate + 1 M sodium chloride (solution B). Under these conditions, stoichiometric HEL-Eα conjugates eluted at ≈6.5% solution B as determined by SDS-PAGE (unpublished data). Stoichiometric HEL-Eα was concentrated through Amicon Ultra 10 kDa MWCO concentrator columns (EMD Millipore), and the concentration was determined from the A$_{280}$ using a predicted 0.1% Cε of 2.116.

## Immunizations and adoptive transfers

NP-CGG or NP-KLH were administered subcutaneously in alum adjuvant (Alhydrogel; Accurate Chemical and Scientific). The antigens were brought to a concentration of 1 mg/ml in PBS and then mixed with an equal volume of alum adjuvant. In most experiments, 25 µl per site were injected into the upper flanks, above the shoulders, and scruff of the neck to generate responses in the draining axillary, brachial, and facial LNs. In some experiments, 25 µl per site were injected into both lower flanks and bilaterally proximal to the base of the tail, generating an immune response in the draining inguinal LNs.

For experiments involving Hy10 B cells, $5 \times 10^4$ HEL-binding naïve follicular B cells were adoptively transferred into congenic (CD45.1 or CD45.2) wild-type mice by intravenous injection into the retro-orbital plexus, 1 d prior to immunization as described (*Allen et al., 2007b*). For *Figure 7D and E*, immunizations were in both ear pinnae with 6.25 µg DEL-OVA/alum, followed by analysis 7 d later of the draining facial lymph nodes. For *Figure 7F and G*, mice were immunized subcutaneously on the left side at the lower flank and base of the tail with 15 µg DEL-OVA/alum. 6.5 d later, mice were challenged with 10 µg HEL-Eα on the left side at the lower flank and base of the tail, followed by analysis 9 hr later of the draining inguinal lymph nodes.

## Ibrutinib treatment of mice

Ibrutinib was synthesized as per published protocols (*Pan et al., 2007*) and dissolved in Captex 355 (ABITEC) at 1.56 mg/mL by sonication. Mice were immunized with NP-CGG subcutaneously as described above, and then starting 3.5 or 6 d after immunization they were injected intraperitoneally twice each day for 3 d with ibrutinib at 12.5 mg/kg dose or vehicle control. Draining lymph nodes were harvested 0.5 d after the last injection.

## Cell culture

Primary B cells were purified from spleens by CD43 and CD11c negative selection as described (*Sullivan et al., 2011*). To enrich for NP-specific B cells from B1-8 mice, Igλ$^+$ B cells were purified by negative selection using biotin-conjugated anti-Igκ (RRID:AB_345328, clone RMK-12, BioLegend) in addition to anti-CD43-biotin (RRID:AB_2255226, clone S7, BD Biosciences) and anti-CD11c-biotin (RRID:AB_313773, clone N418, BioLegend). In order to induce CSR and/or proliferation in vitro, purified splenic B cells or crude splenocytes ($5 \times 10^5$ cells/ml) were cultured in RPMI media containing 10% fetal bovine serum (Life Technologies), 1x penicillin-streptomycin-glutamine (Life Technologies), 50 µM β-mercaptoethanol (Thermo Fisher Scientific), 10 mM HEPES (Life Technologies), 62.5–250 ng/ml rat anti-mouse CD40 antibody (RRID:AB_871724, FGK-45, Miltenyi Biotec), and 25 ng/ml recombinant mouse IL-4 (R and D Systems) in 96-well Microtest U-bottom plates (BD Falcon) with a volume of 200 µl per well. Cells were cultured for 4–5 d in a humidified incubator at 37°C in 5% $CO_2$. Data from entire culture experiments were discarded if viability was unusually poor or positive controls failed to show robust PC differentiation. For treatment of cultured B cells, ibrutinib dissolved in DMSO was added to final concentration of 12.5 nM or as indicated 2 d after initiation of the culture. The final concentration of DMSO in the culture was always 0.025% (w/v). To stimulate B cells with cognate antigen, NP-APC or TNP12-OVA (Biosearch Technologies) was added to a final concentration 0.5 µg/ml on d 3 of culture. The mouse myeloma-derived J558L cell line (RRID:CVCL_3949, obtained from J. Cyster) was maintained in DMEM high glucose media with 10% FBS, 10 mM HEPES, and 1x penicillin-streptomycin-glutamine (Life Technologies) in a humidified incubator at 37°C in 10% $CO_2$. The mouse B lymphoma cell lines BAL 17 (RRID:CVCL_9474, obtained from A. DeFranco), M12 (obtained from J. Cyster), and WEHI-231 (RRID:CVCL_0577, obtained from J. Cyster) were maintained in RPMI media containing 10% fetal bovine serum (Life Technologies), 1x penicillin-streptomycin-glutamine (Life Technologies), 10 mM HEPES (Life Technologies) in a humidified incubator at 37°C in 5% $CO_2$. Anti-IgM stimulation of WEHI-231 cells was with 5 µg/ml F(ab')2 fragment goat anti-mouse IgM, µ chain-specific (RRID:AB_2338474, Jackson ImmunoResearch). Phoenix-Eco cells (RRID:CVCL_H717, obtained from K.M. Ansel, [*Swift et al., 2001*]), which were used for retroviral packaging, were seeded into multi-well plates with DMEM high glucose media containing 10% FBS, 10 mM HEPES, 1x penicillin-streptomycin-glutamine, adjusting the cell density to reach 50–70% confluency at the time of transfection. We confirmed that cell lines had characteristics consistent with published results, by flow cytometric evaluation of surface marker expression and/or

functional assays. As most of our study involved short-term cultures of primary cells isolated from mouse tissues, testing for mycoplasma in cell cultures was not performed.

## Cloning of BCRs and construction of chimeric BCRs

The $V_H$ (designated V4) and $V_\kappa$ genes of the IgG1 hybridoma clone 1B7.11 (M. Wabl lab, UCSF), specific for TNP, were cloned by degenerate PCR as described (*Bradbury et al., 1995*). The coding sequences of membrane IgG1 and secreted IgG1 were amplified from cDNA of the 1B7.11 hybridoma as well. The coding sequences of membrane and secreted isoforms of IgM, IgD, IgA, and IgE were cloned from cDNA prepared from C57BL/6 B cells activated in vitro. The coding sequence of V4 was fused to the constant regions of different BCR isotypes by overlapping PCR. Expression constructs of chimeric BCRs were obtained using the Gibson Assembly Master Mix (New England BioLabs). Expression constructs of BCRs with site-directed mutations were obtained using the Q5 site-directed mutagenesis kit (New England BioLabs). The identity of all expression constructs have been verified by Sanger sequencing.

## Construction of retroviral expression vector

To allow relatively high and constitutive expression of exogenous genes in primary lymphocytes and B cell lines, we modified a self-inactivating retroviral vector pQCXIN (Clontech). The CMV immediate early promoter, the internal ribosome entry site, and the neomycin resistance gene in the vector were replaced with the human EF1α promoter from the pEF/myc/nuc plasmid (Life Technologies). To allow assessment of the expression of BCR constructs by retroviral transduction, the coding sequences of the fluorescent protein Cerulean (*Rizzo et al., 2004*) (obtained from Addgene), light chain, and heavy chains of different isotypes of BCR were linked in-frame by T2A peptide sequences without internal stop codons and cloned downstream of the EF1α promoter using the restriction sites indicated in *Figure 1—figure supplement 1*. The T2A-light chain cassette was only included in the retroviral vectors for expression of TNP-specific BCRs (*Figure 1E*).

## Expression of exogenous BCRs delivered by recombinant retroviral vectors in primary B cells or B cell lines

Phoenix-Eco cells (*Swift et al., 2001*) were transfected with a mixture of 70% retroviral plasmid DNA and 30% MSCV ecotropic gag-pol-env plasmid DNA (from J. Cyster, UCSF) using *Trans*IT -LT1 Transfection Reagent (Mirus Bio) according to the manufacturer's instructions. The medium of the transfected cells were replaced with fresh medium the next morning and again with fresh medium containing 1x ViralBoost (Alstem) in the evening. The next day, the retroviral supernatant was added to primary B cells or B cell lines together with 10 mM HEPES and 5 μg/ml polybrene, followed by centrifugation at 1100x g at room temperature for 90 min. The spinfected cells were then resuspended in the original growth media for further culture. Primary B cells were analyzed by flow cytometry 3 d after spinfection, whereas B cell lines were analyzed at multiple time points as indicated.

## Flow cytometry

Cell suspensions were prepared from LNs and were stained with antibodies (*Supplementary file 1*) essentially as described (*Yang et al., 2012*). With the exception of the apoptosis analyses described below, nonviable cells were excluded by labeling cells during surface staining with the fixable viability dye eFluor780 (eBioscience) as described (*Yang et al., 2012*). Intracellular IgE staining was as described (*Yang et al., 2012*). Briefly, cell surface IgE was first blocked with a large excess of unconjugated anti-IgE antibody RME-1 (RRID:AB_315073, BioLegend) during surface staining, then cells were fixed and permeabilized using the Cytofix/Cytoperm Fixation/Permeabilization solution kit (BD Biosciences). Then intracellular IgE was stained with fluorescently-labeled RME-1 (see *Supplementary file 1*). For the detection of NP-binding cells, APC (ProZyme) was conjugated to the succinimidyl ester of NP (NP-Osu; Biosearch Technologies) at a ratio of 1 mg APC to 80 μg NP-Osu as described (*McHeyzer-Williams and McHeyzer-Williams, 2004*) and then purified on Bio-Spin 6 or Bio-Spin 30 spin columns (Bio-Rad) equilibrated with PBS. For the preparation of HEL-Alexa 647, lyophilized HEL (Sigma-Aldrich) was resuspended at 1.1 mg/mL in DPBS (Life Technologies), and then diluted with 10% v/v 1 M sodium bicarbonate. To this solution, an 8.8-fold molar excess of

Alexa Fluor 647 carboxylic acid, succinimidyl ester (Life Technologies) was added. After incubation at room temperature for two hours with gentle mixing, excess free Alexa Fluor 647 dye was removed by passing the solution through two Bio-Spin 6 columns (Bio-Rad) equilibrated with PBS. Flow cytometry data were collected on an LSR Fortessa (BD) and analyzed with FlowJo v10. All samples were gated on FSC-A versus SSC-A, over a broad range of FSC-A to include blasting lymphocytes, followed by FSC-W versus FSC-H and then SSC-W versus SSC-H gates to exclude doublets. Some two dimensional plots are shown with 'large dots' for better visualization of rare events.

## Apoptosis analysis

For the analysis of apoptosis of cultured primary B cells and B cells lines, annexin V staining and cell surface antigen staining and washing were done in annexin staining buffer (10 mM HEPES (pH7.4), 0.14 M NaCl, 2.5 mM $CaCl_2$). The primary B cells were then fixed and permeabilized for IgE and IgG1 staining. For the ex vivo analysis of apoptosis in GC B cells, after surface staining, fixed and permeabilized cells from draining LNs were stained with anti-activated caspase 3 antibody (RRID: AB_1727414, clone C92-605, BD Biosciences). Alternatively, cell suspensions from draining LNs were incubated with FITC-VAD-FMK (BioVision) at 37°C for 30', washed once with the buffer included in the CaspGLOW kit (BioVision), and then surface stained with annexin V and for other markers, followed by intracellular staining for IgE and total IgG1, as described above. Our FSC-A and SSC-A gating strategy involved a broad FSC-A gate for lymphocytes that included both viable and recently apoptotic cells, but did not include small cell fragments which tended to be autofluorescent and nonspecifically bound to antibodies, which would have confounded analysis.

## Cell cycle analysis

For cell cycle analysis, each mouse was injected intravenously with 1 mg EdU (Life Technologies) in 1xPBS 6.5 d after immunization. After 3.5 hr, draining LNs were collected and single cell suspensions were processed for cell surface staining followed by EdU detection, using the Click-iT plus EdU-Alexa Fluor 647 kit (Life Technologies), according to the manufacturer's protocol, except that 10 million cells, instead of 1 million cells were used for one test. To ensure acquiring sufficient events of IgE$^+$ B cells, 50 million cells from the draining LNs of each mouse were processed. Consequently, intracellular IgE and total IgG1 were stained as described above, but in the Click-iT saponin-based permeabilization and wash buffer. Finally DAPI was added to label DNA to determine the fraction of cells in the G1, S, and G2/M phases of the cell cycle. The final concentration of DAPI was adjusted to achieve a similar fluorescence intensity of the G1 peak and the signal was collected on a linear scale at the LO speed setting to maximize resolution. Doublets were excluded by FSC-W, SSC-W, and DAPI-W measurements.

## VH186.2 sequence analysis

Bulk populations of NP-specific, IgE$^+$ or IgG1$^+$ GC B cells and PCs were first enriched by magnetic bead-based depletion and then sorted on a FACS Aria 3u (BD), as described (*Yang et al., 2012*). Single cells were then sorted into 96-well plates and VH186.2 sequences were amplified by nested PCR, followed by Sanger sequencing, and analyzed as described (*Yang et al., 2012*).

## Microscopy

To quantify the BCR clustering on the surface of J558L cells transduced with IgE or IgG1 by TIRF microscopy, BCRs were visualized by Venus (*Nagai et al., 2002*), a YFP derivative, fused with the cytoplasmic domain of *Cd79a* as described (*Liu et al., 2010a*, *2010b*; *Tolar et al., 2009*). Cells were stained with 10 μM of the hydrophobic membrane dye DiD on ice for five mins, washed and then loaded onto coverslips that had been pre-coated with Poly-L-Lysine. Cells were allowed to adhere for five mins and then imaged live at 37°C. TIRF microscopy images were captured by an Olympus IX-81 microscope supported by ANDOR iXon+ DU-897D electron-multiplying EMCCD camera, a 514 nm laser, a 568 nm laser, Olympus 100 × 1.45 N.A. objective lens and a TIRF port. TIRF microscopy image capture was controlled by Metamorph software (Molecular Devices) and the exposure time was 100 ms for 512 × 512 pixel images. BCR microcluster fluorescence intensity and size were analyzed by Matlab (MathWorks) (*Source code 1*) or Image J (NIH) as described (*Liu et al., 2010a*, *2010b*; *Tolar et al., 2009*).

Single BCR molecule tracking experiments were performed as described (*Liu et al., 2010a*, *2010b*; *Tolar et al., 2009*). In brief, BCRs labeled with Venus were first photobleached with a high power laser for 5–10s, and then imaged by TIRF microscopy. A 100 × 100 pixel sub-region of the electron-multiplying CCD chip with an exposure time of 30 ms per frame was used, the time resolution of which was sufficient to track the single-molecule BCRs as described. Short-range diffusion coefficients and MSD for individual BCR molecule trajectories were processed as described.

### Statistical analysis

GraphPad Prism v6 or v7 were used for statistical analyses, with appropriate tests chosen based on experimental design after consulting the GraphPad Statistics Guide. All tests were two-tailed. In order to achieve sufficient power to discern meaningful differences, experiments were performed with multiple biological replicates and/or multiple times, with details provided in each individual figure legend. The number of samples chosen for each comparison was determined based on past similar experiments or by performing pilot experiments to assess the expected magnitude of differences. Full statistical results with exact p values for all figures are provided in *Supplementary file 2*.

## Acknowledgements

We thank C Cho for technical assistance; E Titus, R Deo, A Rohaim, and D Minor for assistance with chromatography for conjugate purification; J Cyster for comments on the manuscript; M Wabl for providing AID-deficient mice; K Rajewsky for providing B1-8$^{flox}$ mice; J Zikherman for providing Nur77-GFP mice; and KM Ansel, J Cyster, A DeFranco, O Ksionda, R Locksley, J Roose, K Shokat, A Weiss, and J Zikherman for reagents and helpful advice.

## Additional information

### Funding

| Funder | Grant reference number | Author |
| --- | --- | --- |
| Sandler Asthma Basic Research Center | | Zhiyong Yang Marcus J Robinson Christopher DC Allen |
| National Institute of Allergy and Infectious Diseases | F30AI120517 | Geoffrey A Smith |
| Weston Havens Foundation | | Christopher DC Allen |
| Pew Charitable Trusts | | Christopher DC Allen |

The funders had no role in study design, data collection and interpretation, or the decision to submit the work for publication.

### Author contributions

ZY, Conception and design, Acquisition of data, Analysis and interpretation of data, Drafting or revising the article; MJR, Contributed Hy10 experiments, Gave feedback on the manuscript; XC, Performed microscopy experiments and analysis, Contributed to figure preparation for the manuscript; GAS, Contributed to ibrutinib experiments, Gave feedback on the manuscript; JT, Contributed to ibrutinib experiments, provided advice on signaling studies, Gave feedback on the manuscript; WL, Designed and supervised microscopy experiments and analysis, Drafted corresponding sections of the manuscript; CDCA, Conception and design, Analysis and interpretation of data, Drafting or revising the article

### Author ORCIDs

Marcus J Robinson, http://orcid.org/0000-0001-5677-3078
Geoffrey A Smith, http://orcid.org/0000-0002-1638-4219
Wanli Liu, http://orcid.org/0000-0003-0395-2800
Christopher D C Allen, http://orcid.org/0000-0002-1879-9047

## Ethics

Animal experimentation: The care, maintenance, and experimental manipulation of mice followed guidelines established by the Institutional Animal Care and Use Committee of the University of California, San Francisco under approved protocols AN089524 and AN111286.

## Additional files

### Supplementary files

• Supplementary file 1. Table of reagents used in flow cytometry.

• Supplementary file 2. Table of exact p values from all statistical comparisons

• Source code 1. Matlab source code.

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
