## [Decision Letter]

Thank you for submitting your article "Regulation of B Cell Fate by Chronic Activity of the IgE B Cell Receptor" for consideration by *eLife*. Your article has been favorably evaluated by Tadatsugu Taniguchi as the Senior Editor and three reviewers, including Maria Lafaille (Reviewer #3) and a member of our Board of Reviewing Editors.

The reviewers have discussed the reviews with one another and the Reviewing Editor has drafted this decision to help you prepare a revised submission.

Summary:

Overall this manuscript is technically strong and addresses an important issue of current interest in immunology. And, all of the reviewers recognize that, despite recent publication of the similar paper (Haniuda et al; NI), this manuscript contains more extensive information, thereby contributing significantly to the understanding of the mechanisms that determine IgE cell fate determination. However, several concerns are raised by the three reviewers, and they discussed which point should be requested to make conclusions more solid.

Essential revisions:

As mentioned by the authors, the interpretation of the in vivo Figure 5 data is a little bit difficult, because so many processes are going on in generation of GC B cells and eventual PC generation. Requests for detailed analysis in this part in would not be fair, because this is not a major point in this manuscript. But, we request one thing. Because IgE PCs measured in Figure 5 are supposed to include both extrafollicular PCs and GC-derived PCs. So therefore, simple comparison of IgE GC B cells and IgE PCs should be careful, in terms of differentiation of GC cells towards PC fate. To distinguish extrafollicular- versus GC-derived PCs, by using *Blnk* ko vs. wildtype or *Syk* hetero- vs. wildtype, the numbers of IgG1 PCs and IgE PCs which undergo SHM should be measured.

---

## [Author Response]

*Essential revisions:*

*As mentioned by the authors, the interpretation of the in vivo Figure 5 data is a little bit difficult, because so many processes are going on in generation of GC B cells and eventual PC generation. Requests for detailed analysis in this part in would not be fair, because this is not a major point in this manuscript. But, we request one thing. Because IgE PCs measured in Figure 5 are supposed to include both extrafollicular PCs and GC-derived PCs. So therefore, simple comparison of IgE GC B cells and IgE PCs should be careful, in terms of differentiation of GC cells towards PC fate. To distinguish extrafollicular- versus GC-derived PCs, by using Blnk ko vs. wildtype or Syk hetero- vs. wildtype, the numbers of IgG1 PCs and IgE PCs which undergo SHM should be measured.*

We appreciate the suggestion by the reviewers to determine the extent to which PCs may be derived from extrafollicular versus GC pathways by SHM analysis. We performed this analysis with conditional *Syk* heterozygous versus control mice as suggested. The data are now included as Figure 5—figure supplement 2. We did this analysis at 9 days after immunization, consistent with all of the in vivo analyses in Figure 5, which had shown a consistent increase in IgE^+^ GC B cells and variable increases in IgE^+^ PCs under various BCR signaling perturbations. Interestingly, at this timepoint, more than 90% of IgE^+^ and IgG1^+^ PCs had germline sequences, suggesting they were from the extrafollicular pathway. However, a substantial fraction of GC B cells also had germline sequences, leaving open the possibility that some fraction of the PCs were GC-derived. In addition, some increase in the frequency of PCs with somatic mutations was observed in the conditional *Syk* heterozygous mice. The implication of these findings is that BCR signaling perturbations affect IgE responses in both the extrafollicular and GC pathways. We have revised the text of the manuscript accordingly.